# Efficient Distributionally Robust Assortment Optimization in MNL Bandits

**Yunfan Zhang** [1]   **Yuxuan Han** [1]   **Zhengyuan Zhou** [1]

## Abstract

We investigate the distributionally robust assortment optimization (DRAO) problem under the contextual multinomial logit (MNL) choice model, where the decision-maker seeks to maximize revenue against worst-case distributional deviations. To address potential distribution shifts relative to the observed data environment, we study DRAO under ambiguity sets defined by three divergences: total variation (TV), Kullback–Leibler (KL), and chi-square ($\chi^2$). Incorporating robust concerns poses challenges for both algorithm design and theoretical analysis. By leveraging strong duality results from the distributionally robust optimization literature and integrating them into the assortment optimization procedures, we develop tailored polynomial-time algorithms under each divergence. We further provide a theoretical analysis and establish high-probability finite-sample bounds on the robust sub-optimality gap for all three robust formulations.

## 1. Introduction

Choice modeling and the associated assortment optimization problem are central topics in decision making. In a typical assortment problem, a seller managing a large inventory of $N$ items must select a subset $S$ of products (an assortment), often subject to a cardinality constraint $|S| \leq K$, to display to an arriving customer. The seller's objective is to maximize expected profit based on the customer's resulting choice behavior. While this is a combinatorial optimization problem searching over all possible $K$-sized assortments in general, the multinomial logit (MNL) choice model admits polynomial-time procedures for computing optimal assortments (Rusmevichientong et al., 2010; Davis et al., 2013; Avadhanula et al., 2016). This computational tractability,

together with its empirical success, has made MNL model or its contextual variant a popular model of study.

In modern assortment optimization, model uncertainty is unavoidable: the deployed choice behavior may deviate from the assumed model due to misspecification, and the customer preference may shift relative to the historical data used for model calibration. As a result, optimizing against a single nominal model can lead to fragile assortments and degraded out-of-sample performance. This has motivated a growing literature on robust assortment optimization (Rusmevichientong & Topaloglu, 2012; Li & Ke, 2019; Jin et al., 2022; Désir et al., 2024; Wang et al., 2024b). Among various robust formulations, the distributionally robust assortment optimization(DRAO) formulation is particularly well-suited because the objective is inherently an expectation under a choice distribution; robustifying over such choice distribution therefore naturally amounts to guarding against distributional deviations in customer choice.

Despite the growing interest in robust assortment optimization, the literature on DRAO under MNL model remains limited. To the best of our knowledge, only two recent works study DRAO in this setting, both focusing on ambiguity sets defined by the Kullback–Leibler (KL) divergence (Jin et al., 2022; Lu et al., 2026). Specifically, Jin et al. (2022) considers a known-model setting and focus on developing computationally tractable algorithms, while Lu et al. (2026) builds on this framework to establish finite-sample guarantees in the data-driven non-contextual MNL setting. Despite this progress, several gaps remain open. First, even in the known-model setting considered by Jin et al (2022), it is unclear how to design computationally efficient robust-assortment solvers for ambiguity sets beyond KL. Second, existing results largely focus on non-contextual models; extending DRAO to contextual MNL, where choice probabilities depend on features, remains unexplored.

In this work, we study data-driven distributionally robust assortment optimization under the linear contextual MNL model, using ambiguity sets defined by TV, KL, and Chi-square distances. We develop computationally tractable algorithms for robust assortment selection and establish finite-sample guarantees for learning robust optimal assortments from data. Our main contributions are summarized in Table 1 and below.

---
[1]Stern School of Business, New York University, New York, NY, USA. Correspondence to: Zhengyuan Zhou <zzhou@stern.nyu.edu>.

*Proceedings of the 43$^{rd}$ International Conference on Machine Learning*, Seoul, South Korea. PMLR 306, 2026. Copyright 2026 by the author(s).

### 1.1. Our Contributions

**Efficient Algorithms for Known-Model DRAO.** We first consider DRAO in the *known-model* MNL setting under TV, KL, and Chi-square ambiguity sets. While strong duality reduces the inner robust minimization to a univariate dual problem, the resulting outer problem couples a continuous dual variable with combinatorial assortment search. The exploitable structure of this coupling differs sharply across divergences, so identifying divergence-specific polynomial-time solvers is the main algorithmic challenge. For the TV ambiguity set, we exploit the piecewise-linear structure of the dual objective and obtain a novel exact algorithm with $O(N^2 \log N)$ runtime. For the Chi-square ambiguity set, we propose an efficient Golden-Section Search procedure that yields an $\varepsilon$-optimal solution in $O(N \log N \cdot \log(1/\varepsilon))$ time. To the best of our knowledge, these efficient solvers for TV and Chi-square robust assortment are new even in the static, known-parameter setting. Finally, we include an $\epsilon$-optimal algorithm for the KL ambiguity set for completeness, which aligns our framework with prior work (Jin et al., 2022; Lu et al., 2026).

**Finite-Sample Guarantees for Data-Driven Contextual DRAO.** We then study the *linear contextual* MNL setting where $\boldsymbol{\theta}^\star$ is unknown, a setting that has been extensively studied in the non-robust literature (Oh & Iyengar, 2021; Lee & Oh, 2024; 2025; Han et al., 2025a; Dong et al., 2023). Our approach uses a $\lambda$-regularized MLE to construct lower-confidence (LCB) attractions, then applies the algorithms above to the LCB model to obtain a robust assortment $\hat{S}_D$. Establishing high-probability suboptimality bounds for $\hat{S}_D$ is not a direct corollary of either non-robust contextual analyses (Han et al., 2025a) or non-contextual robust analyses (Lu et al., 2026); we address this via a unified two-term decomposition whose second term is non-positive at the robust optimum due to a monotonicity property (details in Section 4). We derive rates of $\tilde{O}\left(\sum_{i \in S^\star_{\mathrm{TV}}} p(i|S^\star_{\mathrm{TV}}) \|\boldsymbol{x}_i\|_{\boldsymbol{H}_T^{-1}(\boldsymbol{\theta}^\star)}\right)$ for the TV distance, $\tilde{O}_\delta\left(\sum_{i \in S^\star_{\mathrm{KL}}} p(i|S^\star_{\mathrm{KL}}) \|\boldsymbol{x}_i\|_{\boldsymbol{H}_T^{-1}(\boldsymbol{\theta}^\star)}\right)$ for the KL distance, and $\tilde{O}_\delta\left(\sqrt{\sum_{i \in S^\star_{\chi^2}} p(i|S^\star_{\chi^2}) \|\boldsymbol{x}_i\|_{\boldsymbol{H}_T^{-1}(\boldsymbol{\theta}^\star)}}\right)$ for the Chi-squared distance, as summarized in Table 1. The probability-weighted self-normalized form yields a feature-based, localized notion of offline coverage that improves upon the assortment-wise condition required by Dong et al. (2023). We complement these guarantees with synthetic experiments validating algorithm scaling, fragility of non-robust policies under distribution shift, and robustness to non-MNL model misspecification (Appendix D).

### 1.2. Related Works

**Distributionally Robust Decision Making.** Incorporating distribution-based worst-case considerations into decision-making has been extensively studied in areas such as contextual bandits and reinforcement learning (e.g., Si et al. (2023); Wang et al. (2024a); Liu et al. (2022); Lu et al. (2024); Chen et al. (2025)). However, this line of work cannot be directly applied to the MNL setting due to the combinatorial nature of the action space. To the best of our knowledge, the only works that study DRO formulations under MNL are Lu et al. (2026); Jin et al. (2022), as discussed above. It is also worth noting that, beyond DRO, other robust approaches have also been studied for assortment optimization with possibly more general choice models (Rusmevichientong & Topaloglu, 2012; Li & Ke, 2019; Wang et al., 2024b; Désir et al., 2024).

**Data-Driven Assortment Optimization.** Data-driven assortment optimization under the MNL model has been extensively studied in both the online setting (Agrawal et al., 2019; Lee & Oh, 2024; 2025; Agrawal et al., 2023; Perivier & Goyal, 2022; Oh & Iyengar, 2021) and the offline setting (Dong et al., 2023; Han et al., 2025b;a). Among these, the most closely related works are Dong et al. (2023); Han et al. (2025b;a), which also study offline assortment optimization under possibly partial coverage, but focus on the non-robust setting. Our bounds share a similar spirit to Han et al. (2025a), where a feature-based coverage condition is introduced to control the suboptimality gap.

**Conflict of Interest Disclosure.** The authors have no financial conflicts of interest to disclose.

**Notations.** For a positive integer $N$, we denote the set of indices as $[N] := \{1, 2, \ldots, N\}$. We use bold uppercase letters (e.g., $\boldsymbol{X}$) and bold lowercase letters (e.g., $\boldsymbol{v}$) to denote matrices and vectors, respectively. $\Delta(S)$ represents the probability simplex over the set $S$. For any real number $a \in \mathbb{R}$, we define its positive part as $a_+ := \max\{a, 0\}$. We use $O(\cdot)$ to denote standard asymptotic boundedness, $\tilde{O}(\cdot)$ to suppress polylogarithmic factors, and $\tilde{O}_\delta(\cdot)$ to suppress factors depending on $\delta$.

## 2. Preliminary

### 2.1. Problem Formulation

**MNL Choice Model.** We consider an assortment optimization problem with $N$ items, indexed by $[N] = \{1, \ldots, N\}$. There is a *seller* and a *customer* in this interaction. The seller offers an assortment $S \in \mathcal{S}_K$, where $\mathcal{S}_K = \{S \subseteq [N] : |S| \leq K\}$ represents the set of feasible assortments with cardinality at most $K$. The customer then makes a choice $c \in S_+ := S \cup \{0\}$, where $0$ represents

| Works | Leading-Order Sub-Optimality | Contextual | Robust | Distance |
|---|---|---|---|---|
| Han et al. (2025b) | $\tilde{O}\left(\frac{K}{\sqrt{\min_{i \in S^\star} n_i}}\right)$ | ✗ | ✗ | – |
| Lu et al. (2026) | $\tilde{O}_\delta\left(\frac{K}{\sqrt{\min_{i \in S^\star_{\text{KL}}} n_i}}\right)$ | ✗ | ✓ | KL |
| Dong et al. (2023) | $\tilde{O}\left(\sqrt{\frac{N}{n_{S^\star} \cdot \min_{i \in [N], |S|=K} p(i|S)}}\right)$ | ✓ | ✗ | – |
| Han et al. (2025a) | $\tilde{O}\left(\sqrt{\sum_{i \in S^\star} p(i|S^\star)p(0|S^\star)\|x_i\|^2_{\boldsymbol{H}_T^{-1}(\boldsymbol{\theta}^\star)}}\right)$ | ✓ | ✗ | – |
| **This work** | $\tilde{O}\left(\sum_{i \in S^\star_{\text{TV}}} p(i|S^\star_{\text{TV}})\|\boldsymbol{x}_i\|_{\boldsymbol{H}_T^{-1}(\boldsymbol{\theta}^\star)}\right)$ | ✓ | ✓ | TV |
| **This work** | $\tilde{O}_\delta\left(\sum_{i \in S^\star_{\text{KL}}} p(i|S^\star_{\text{KL}})\|\boldsymbol{x}_i\|_{\boldsymbol{H}_T^{-1}(\boldsymbol{\theta}^\star)}\right)$ | ✓ | ✓ | KL |
| **This work** | $\tilde{O}_\delta\left(\sqrt{\sum_{i \in S^\star_{\chi^2}} p(i|S^\star_{\chi^2})\|\boldsymbol{x}_i\|_{\boldsymbol{H}_T^{-1}(\boldsymbol{\theta}^\star)}}\right)$ | ✓ | ✓ | $\chi^2$ |

*Table 1.* Comparison of offline data-driven assortment optimization results. Only leading-order terms are presented for clarity. Notation: $S^\star$ and $S^\star_D$ denote the nominal (non-robust) and robust optimal assortments, respectively; $p(i|S)$ is the choice probability of item $i$ under parameter $\boldsymbol{\theta}^\star$; $n_i$ and $n_S$ represent the offline occurrence counts of item $i$ and assortment $S$; $K$ is the assortment capacity; and $\boldsymbol{H}_T(\boldsymbol{\theta}^\star)$ denotes the Hessian matrix. The notation $\tilde{O}_\delta(\cdot)$ suppresses multiplicative factors depending on the uncertainty radius $\delta$.

the "no-purchase" option. Each item has an attraction value $v_i \geq 0$, and we normalize the no-purchase attraction as $v_0 = 1$. Denote attraction vector as $\boldsymbol{v} = (v_0, v_1, ..., v_N)$. Under the multinomial logit (MNL) model, the choice probability for any $i \in S_+$ is

$$p(i|S, \boldsymbol{v}) := \frac{v_i}{\sum_{i \in S_+} v_i} = \frac{v_i}{1 + \sum_{i \in S} v_i}.$$

If the customer purchases item $i \in [N]$, the seller receives a revenue $r_i \in [0, 1]$, while no-purchase yields $r_0 = 0$. Then, the expected revenue of offering assortment $S$ is

$$R(S|\boldsymbol{v}) := \sum_{i \in S_+} r_i p(i|S, \boldsymbol{v}) = \frac{\sum_{i \in S} r_i v_i}{1 + \sum_{i \in S} v_i}. \quad (1)$$

**Distributionally Robust Assortment Optimization** Fix an assortment $S$, to model potential distribution shift relative to the nominal choice model $p(\cdot|S, \boldsymbol{v})$, we consider the distributionally robust variant of (1), first proposed by Rusmevichientong & Topaloglu (2012). Let $\Delta(S_+)$ denote the probability simplex over $S_+$. For any $q \in \Delta(S_+)$, define the expected revenue under $q$ as

$$R(S; q) := \sum_{i \in S_+} r_i q(i).$$

Given a divergence $D(\cdot, \cdot)$ on $\Delta(S_+)$ and a radius $\delta \geq 0$, we define the ambiguity set centered at a nominal parameter $\boldsymbol{v}$ as

$$\mathcal{P}_D(\boldsymbol{v}, \delta, S) := \{q \in \Delta(S_+), D(q, p(\cdot|S, \boldsymbol{v})) \leq \delta\}. \quad (2)$$

The worst-case revenue under $\mathcal{P}_D(\boldsymbol{v}, \delta, S)$ is defined as

$$R_D^\delta(S|\boldsymbol{v}) := \min_{q \in \mathcal{P}_D(\boldsymbol{v}, \delta, S)} R(S; q),$$

The corresponding distributionally robust assortment optimization problem is

$$R_D^\star := \max_{S \in \mathcal{S}_K} R_D^\delta(S|\boldsymbol{v}). \quad (3)$$

Specifically, we focus on three distinct $f$-divergences. For any $p, q \in \Delta(S_+)$:
**(i) Total variation distance:**

$$D_{\text{TV}}(q, p) = \frac{1}{2}\|q - p\|_1.$$

**(ii) Kullback–Leibler distance:**

$$D_{\text{KL}}(q, p) = \sum_{i \in S_+} q(i) \log\left(\frac{q(i)}{p(i)}\right).$$

**(iii) Chi-square distance:**

$$D_{\chi^2}(q, p) = \sum_{i \in S_+} \frac{(q(i) - p(i))^2}{p(i)}.$$

### 2.2. Contextual MNL and Data-Driven DRAO

**Linear Contextual MNL.** In this paper, we further assume the attraction values are parameterized by item features. Specifically, each item $i$ is associated with a feature vector $\boldsymbol{x}_i \in \mathbb{R}^d$, and there exists an unknown parameter $\boldsymbol{\theta}^\star \in \mathbb{R}^d$ such that

$$v_i = \exp(\boldsymbol{x}_i^\top \boldsymbol{\theta}^\star).$$

We set $\boldsymbol{x}_0 = \boldsymbol{0}$ for the no-purchase option. Let $\boldsymbol{X} = (\boldsymbol{x}_0, \boldsymbol{x}_1, ..., \boldsymbol{x}_N) \in \mathbb{R}^{d \times (N+1)}$ denote the full feature collection. Then, given an assortment $S$, the contextual MNL

choice probability for any $i \in S_+$ is defined as

$$p(i|S, \boldsymbol{\theta}^\star) := \frac{\exp(\boldsymbol{x}_i^\top \boldsymbol{\theta}^\star)}{1 + \sum_{j \in S} \exp(\boldsymbol{x}_j^\top \boldsymbol{\theta}^\star)}.$$

The corresponding expected revenue is given by

$$R(S|\boldsymbol{\theta}^\star) = \frac{\sum_{i \in S} r_i \exp(\boldsymbol{x}_i^\top \boldsymbol{\theta}^\star)}{1 + \sum_{j \in S} \exp(\boldsymbol{x}_j^\top \boldsymbol{\theta}^\star)}.$$

Throughout this paper, we use the following standard boundedness assumption on the parameters.

**Assumption 2.1** (Boundedness). For all $i \in [N]$, the feature vectors satisfy $\|\boldsymbol{x}_i\|_2 \leq 1$. The unknown parameter vector satisfies $\|\boldsymbol{\theta}^\star\|_2 \leq W$ for some known constant $W > 0$.

**Data-Driven Robust Assortment Optimization.** We study a data-driven version of (3) in a linear contextual MNL model with $\boldsymbol{X}, \boldsymbol{\theta}^\star$, where the seller does not know the underlying parameter $\boldsymbol{\theta}^\star$, and thus does not know the nominal choice distribution exactly. The seller observes a historical dataset $\{(c_t, \boldsymbol{X}_t, S_t)\}_{t=1}^T$, where at round $t$ an assortment $S_t \in \mathcal{S}_K$ is offered under features $\boldsymbol{X}_t$, and the realized choice is $c_t \in (S_t)_+$. Given a target feature matrix $\boldsymbol{X}$ and a revenue vector $\boldsymbol{r} = (r_0, r_1, ..., r_N)$ (typically $r_0 = 0$), the worst-case revenue under the ambiguity set $\mathcal{P}_D(\boldsymbol{\theta}^\star, \delta, S) := \{q \in \Delta(S_+), D(q, p(\cdot|S, \boldsymbol{\theta}^\star)) \leq \delta\}$ is defined as

$$R_D^\delta(S|\boldsymbol{\theta}^\star) := \min_{q \in \mathcal{P}_D(\boldsymbol{\theta}^\star, \delta, S)} R(S; q).$$

Our objective is to identify an optimal robust assortment $S_D^\star \in \mathcal{S}_K$ that maximizes the worst-case revenue:

$$S_D^\star := \operatorname*{argmax}_{S \in \mathcal{S}_K} R_D^\delta(S|\boldsymbol{\theta}^\star). \tag{4}$$

This objective can be equivalently analyzed by minimizing the sub-optimality gap, defined for any $S \in \mathcal{S}_K$ as

$$\mathrm{SubOpt}_D^\delta(S|\boldsymbol{\theta}^\star) := R_D^\delta(S_D^\star|\boldsymbol{\theta}^\star) - R_D^\delta(S|\boldsymbol{\theta}^\star).$$

## 3. Efficient Algorithms for DRAO

In this section, we address the distributionally robust assortment optimization (DRAO) problem assuming the attraction vector $\boldsymbol{v}$ is known. Consequently, for any assortment $S$, the nominal choice probabilities $p(\cdot|S, \boldsymbol{v})$ are fully determined. Our objective is to solve the following max-min problem

$$R_D^\star = \max_{S \in \mathcal{S}_K} \min_{q \in \mathcal{P}_D(\boldsymbol{v}, \delta, S)} R(S; q).$$

This formulation is computationally intractable due to the non-convex nature of the inner minimization over the ambiguity set. To address this, we leverage strong duality theory to reformulate the max-min problem into a tractable univariate optimization, effectively decoupling the distributional uncertainty from the combinatorial assortment search.

### 3.1. DRAO under TV Ambiguity Set

We begin by analyzing the DRAO problem under the Total Variation ambiguity set. By exploiting the convex conjugate structure of the robust objective, we establish the following strong duality result, which reduces the continuous robust optimization to a discrete search.

**Lemma 3.1.** *Given an assortment $S \in \mathcal{S}_K$ and an attraction vector $\boldsymbol{v}$, the robust revenue under TV ambiguity set satisfies:*

$$R_{\mathrm{TV}}^\delta(S|\boldsymbol{v})$$
$$= -\min_{\alpha \in [0, 1/\delta]} \left\{ \mathbb{E}_{i \sim p(\cdot|S, \boldsymbol{v})} \left[ (\alpha - r_i)_+ \right] - (1 - \delta)\alpha \right\} \tag{5}$$
$$= \max_{\alpha \in [0, 1/\delta]} \Phi(\alpha; S, \boldsymbol{v}) = \max_{i \in [N]} \Phi(r_i; S, \boldsymbol{v}), \tag{6}$$

*where $\Phi(\alpha; S, \boldsymbol{v}) := \mathbb{E}_{i \sim p(\cdot|S, \boldsymbol{v})} [\min\{\alpha, r_i\}] - \delta\alpha$.*

*Remark* 3.2. The dual formulation in (5) adapts the result from Iyengar (2005) (see also Lemma 5 in Panaganti et al. (2022)) to our setting; we rely on this continuous dual form for our regret analysis. Equation (6) presents our key algorithmic insight: the dual objective function $\Phi(\alpha; S, \boldsymbol{v})$ is concave and piecewise linear, with "knots" at the reward values $r_i$. Consequently, the optimal dual variable $\alpha^*$ must coincide with one of the item revenues. Specifically, the maximum is achieved at $r_k$, where $k$ corresponds to the $(1-\delta)$-quantile of the reward distribution (see Appendix A.1 for details).

**Algorithm Design.** In the standard non-robust setting, the optimal assortment under a cardinality constraint $K$ can be computed efficiently using the StaticMNL (See Algorithm 6 in Appendix B.1) introduced by Rusmevichientong et al. (2010). We leverage this result to construct our robust optimization procedure.

Based on Lemma 3.1, our objective is to find the pair $(S_{\mathrm{TV}}^\star, \alpha^\star)$ that maximizes the dual function. Since $\alpha^\star \in \{r_i\}_{i \in [N]}$, we can interchange the maximization order:

$$\max_{S \in \mathcal{S}_K} \max_{i \in [N]} \Phi(r_i; S, \boldsymbol{v}) = \max_{i \in [N]} \max_{S \in \mathcal{S}_K} \Phi(r_i; S, \boldsymbol{v})$$

Accordingly, we design the algorithm that iterates through each candidate dual variable $\alpha \in \{r_1, \ldots, r_N\}$. For a fixed $\alpha$, the term $-\delta\alpha$ is constant; thus, maximizing $\Phi(\alpha; S, \boldsymbol{v})$ is equivalent to maximizing $\mathbb{E}_{i \sim p(\cdot|S, \boldsymbol{v})} [\min\{\alpha, r_i\}]$. That is, we solve a standard MNL assortment problem in which each item's effective revenue is truncated to $\bar{r}_i = \min\{\alpha, r_i\}$. This subproblem is exactly solved by StaticMNL using these truncated revenues, as shown in Algorithm 1.

**Computational Complexity.** Since the StaticMNL subroutine runs in $O(N \log N)$ time and is invoked $N$ times, the total computational complexity of our approach is $O(N^2 \log N)$. We empirically verify this scaling in Appendix D.1.

**Algorithm 1** TV-Robust MNL

1: **Input:** Revenue $r$, attraction value $v$, uncertainty radius $\delta$, capacity $K$.
2: **Initialize:** $S_{\text{TV}} \leftarrow \emptyset$, $R_{\text{TV}} \leftarrow 0$.
3: **for** each $\alpha \in \{r_1, \ldots, r_N\}$ **do**
4:    $\bar{r}_i \leftarrow \min\{\alpha, r_i\}$ for all $i \in [N]$.
5:    $S(\alpha) \leftarrow \texttt{StaticMNL}(\bar{r}_i, v, \mathcal{S}_K)$ by Algorithm 6.
6:    $\Phi(\alpha) \leftarrow \Phi(\alpha; S(\alpha), v)$.
7:    **if** $\Phi(\alpha) > R_{\text{TV}}$ **then**
8:       $S_{\text{TV}} \leftarrow S(\alpha)$, $R_{\text{TV}} \leftarrow \Phi(\alpha)$.
9:    **end if**
10: **end for**
11: **Return** $S_{\text{TV}}^\star \leftarrow S_{\text{TV}}$.

## 3.2. DRAO under KL Ambiguity Set

We next study the DRAO problem under the Kullback-Leibler ambiguity set. By applying Theorem 1 in Hu & Hong (2013), we reformulate the worst-case revenue as a univariate optimization problem.

**Lemma 3.3** (Theorem 1 in Hu & Hong (2013))**.** *Given an assortment $S \in \mathcal{S}_K$ and an attraction vector $v$, the robust revenue under KL ambiguity set satisfies:*

$$R_{\text{KL}}^\delta(S|v) = -\min_{\rho \geq 0} \left\{ \rho \log \left( \mathbb{E}_{i \sim p(\cdot|S,v)}[\exp(-r_i/\rho)] \right) + \rho\delta \right\}$$
$$= \max_{0 \leq \rho \leq 1/\delta} G(\rho; S, v), \quad (7)$$

*where* $G(\rho; S, v) := -\rho \log \left( \mathbb{E}_{i \sim p(\cdot|S,v)}[\exp(-r_i/\rho)] \right) - \rho\delta$.

**Algorithm Design.** The KL ambiguity set leads to a more intricate robust objective: the dual reformulation involves log–sum–exp terms and a continuous dual variable. Unlike the TV case, we cannot restrict our search to a finite set of candidates. Inspired by the methods in Jin et al. (2022); Lu et al. (2026), we design an algorithm that computes an $\varepsilon$-optimal solution for:

$$(S_{\text{KL}}^\star, \rho^\star) := \operatorname*{argmax}_{S \in \mathcal{S}_K, 0 \leq \rho \leq 1/\delta} G(\rho; S, v).$$

**Step 1: Root-finding via Bisection.** By introducing an auxiliary variable $t \in [0, 1]$, we prove in Lemma B.1 (see Appendix B.2) that the optimal KL-robust revenue $R_{\text{KL}}^\star$ is the unique root of a monotone function:

$$R_{\text{KL}}^\star = \max_{0 \leq t \leq 1} \{\mathcal{G}(t) \leq 0\}. \quad (8)$$

Here, $\mathcal{G}(t)$ is defined as

$$\mathcal{G}(t) := \min_{S \subseteq \mathcal{N}(t) \cap \mathcal{S}_K, \rho \in [0, 1/\delta]} \sum_{i \in S_+} g_i(t, \rho),$$

with $g_i(t, \rho) := v_i (\exp((t - r_i)/\rho) - \exp(-\delta))$ and $\mathcal{N}(t) := \{i \in [N] : t \leq r_i\}$. Importantly, $\mathcal{G}(t)$ is monotone increasing in $t$. Therefore, (8) can be efficiently solved by a bisection search if we can efficiently evaluate $\mathcal{G}(t)$ for any fixed $t$; see Algorithm 2.

**Algorithm 2** KL-Robust MNL

1: **Input:** Error tolerance $\varepsilon_t, \varepsilon > 0$.
2: **Initialize:** $t_L \leftarrow 0$, $t_R \leftarrow 1$.
3: **while** $t_R - t_L > \varepsilon_t$ **do**
4:    $t \leftarrow (t_L + t_R)/2$.
5:    $(S(t), \mathcal{G}(t)) \leftarrow \texttt{EVAL-G}(t, \varepsilon)$ by Algorithm 3.
6:    **if** $\mathcal{G}(t) > \varepsilon$ **then**
7:       $t_R \leftarrow t$.
8:    **else if** $\mathcal{G}(t) < 0$ **then**
9:       $t_L \leftarrow t$, $S_L \leftarrow S(t)$.
10:    **end if**
11: **end while**
12: **Return:** $S_{\text{KL}}^\varepsilon \leftarrow S_L$.

**Step 2: Evaluating $\mathcal{G}(t)$.** The evaluation of $\mathcal{G}(t)$ is nontrivial because the optimal assortment $S$ may change with $\rho$. We exploit the geometric structure of the cost curves $\{g_i(t, \cdot)\}_{i \in \mathcal{N}(t)}$. As established in Lemma B.2 (see Appendix B.2), any two curves $g_i(t, \cdot)$ and $g_j(t, \cdot)$ intersect at most once over $\rho \in [0, 1/\delta]$. This implies that the order of the cost functions is fixed between consecutive intersection points, thus evaluating $\mathcal{G}(t)$ reduces to a finite collection (at most $O(N^2)$) of one-dimensional minimizations in $\rho$. Furthermore, Lemma B.2 shows that the objective $\sum_{i \in S_+} g_i(t, \rho)$ is quasi-convex in $\rho$. Thus, we can efficiently find the local minimum within each interval. The Algorithm 3 elaborates this procedure.

**Algorithm 3** $\texttt{EVAL-G}(t)$: Evaluation of $\mathcal{G}(t)$

1: **Input:** Revenue $r$, attraction value $v$, uncertainty radius $\delta$, capacity $K$, $t \in [0, 1]$, error tolerance $\varepsilon > 0$.
2: $\mathcal{N}(t) \leftarrow \{i \in [N] : t \leq r_i\}$.
3: Compute intersection points of $\{g_i(t, \rho)\}_{i \in \mathcal{N}(t)}$ over $[0, 1/\delta]$. Sort them as $0 = \rho_0 < \rho_1 < \cdots < \rho_m = 1/\delta$.
4: **for** $\ell = 1, \ldots, m$ **do**
5:    Pick an arbitrary $\rho' \in (\rho_{\ell-1}, \rho_\ell)$.
6:    Let $S_\ell$ be the set of up to $K$ items in $\mathcal{N}(t)$ with the smallest values of negative $g_i(t, \rho')$.
7:    Compute $\hat{G}_\ell$ as an $\varepsilon$-optimal value of $\min_{\rho_{\ell-1} \leq \rho \leq \rho_\ell} \sum_{i \in (S_\ell)_+} g_i(t, \rho)$ via bisection search.
8: **end for**
9: $\hat{\ell} \leftarrow \operatorname{argmin}_{\ell \in [m]} \hat{G}_\ell$, $G \leftarrow \hat{G}_{\hat{\ell}}$, $S \leftarrow S_{\hat{\ell}}$.
10: **Output:** $(S, G)$.

**Computational Complexity.** The outer loop of Algorithm 2 performs a bisection search on $t$, requiring

$O(\log(1/\varepsilon_t))$ iterations to converge. Within each iteration, the subroutine EVAL-G (Algorithm 3) executes in $O(N^2 \log(1/(\delta\varepsilon)))$ time, a result formally established in Lu et al. (2026). Consequently, the total computational complexity of Algorithm 2 is $\widetilde{O}(N^2)$. We empirically verify this scaling in Appendix D.1.

### 3.3. DRAO under Chi-square Ambiguity Set

Finally, we address the DRAO problem constrained by the Chi-square ambiguity set. By applying the strong duality result from Iyengar (2005); Xu et al. (2023), we reformulate the robust revenue into a tractable univariate optimization problem.

**Lemma 3.4** (Lemma 9 in Xu et al. (2023)). *Given an assortment $S \in \mathcal{S}_K$ and an attraction value $\boldsymbol{v}$, the robust revenue under Chi-square uncertainty set satisfies:*

$$R_{\chi^2}^{\delta}(S|\boldsymbol{v}) = -\min_{\eta \in \mathbb{R}}\left\{\sqrt{(1+\delta)\mathbb{E}_{i \sim p(\cdot|S,\boldsymbol{v})}[(\eta - r_i)^2]} - \eta\right\}$$

$$= \max_{0 \le \eta \le \frac{\sqrt{1+\delta}}{\sqrt{1+\delta}-1}} H(\eta; S, \boldsymbol{v}), \qquad (9)$$

*where $H(\eta; S, \boldsymbol{v}) := \eta - \sqrt{(1+\delta)\mathbb{E}_{i \sim p(\cdot|S,\boldsymbol{v})}[(\eta - r_i)^2]}$.*

**Algorithm Design.** Based on Lemma 3.4, we can interchange the maximization order and seek the pair $(S_{\chi^2}^{\star}, \eta^{\star})$ that maximizes

$$\max_{S \in \mathcal{S}_K} \max_{0 \le \eta \le \frac{\sqrt{1+\delta}}{\sqrt{1+\delta}-1}} H(\eta; S, \boldsymbol{v}) = \max_{0 \le \eta \le \frac{\sqrt{1+\delta}}{\sqrt{1+\delta}-1}} \max_{S \in \mathcal{S}_K} H(\eta; S, \boldsymbol{v}).$$

Similar to the KL case, the dual reformulation introduces a continuous scalar variable $\eta$, making exact enumeration impossible. We therefore design an algorithm that returns an $\epsilon$-optimal solution via a one-dimensional search over $\eta$.

**Step 1: Inner Subproblem (Exact Solution).** First, consider the case where the dual variable $\eta$ is fixed. We seek to find the assortment $S$ that maximizes the objective $H(\eta; S, \boldsymbol{v})$. Lemma B.3 (see Appendix B.3) shows that the optimal $\chi^2$-robust revenue $R_{\chi^2}^{\star}$ can be expressed as

$$R_{\chi^2}^{\star} = \max_{0 \le \eta \le \frac{\sqrt{1+\delta}}{\sqrt{1+\delta}-1}} \max_{S \in \mathcal{I}(\eta) \cap \mathcal{S}_K} H(\eta; S, \boldsymbol{v})$$

$$= \max_{0 \le \eta \le \frac{\sqrt{1+\delta}}{\sqrt{1+\delta}-1}} \left\{\eta - \sqrt{(1+\delta)}\sqrt{\eta^2 - \tilde{R}(\eta)}\right\}, \quad (10)$$

where $\tilde{r}_i := 2\eta r_i - r_i^2$, $\tilde{r}_0 := 0$, and $\mathcal{I}(\eta) := \{i \in [N] : \tilde{r}_i \ge 0\}$, $\tilde{R}(\eta) := \max_{S \in \mathcal{I}(\eta) \cap \mathcal{S}_K} \mathbb{E}_{i \sim p(\cdot|S,\boldsymbol{v})}[\tilde{r}_i]$.

For any fixed $\eta$, the inner maximization in (10) reduces to a standard static MNL assortment problem with transformed revenues $\tilde{r}_i$ and candidate set $\mathcal{I}(\eta)$; hence it can be solved exactly and efficiently by StaticMNL (See Algorithm 6 in Appendix B.1).

---

**Algorithm 4** EVAL-H$(\eta)$: Evaluation of $H(\eta)$

1: **Input:** Revenue $\boldsymbol{r}$, attraction value $\boldsymbol{v}$, uncertainty radius $\delta$, capacity $K$, $\eta \in [0, \frac{\sqrt{1+\delta}}{\sqrt{1+\delta}-1}]$.
2: $\tilde{r}_i \leftarrow 2\eta r_i - r_i^2$ for all $i \in [N]$.
3: $\mathcal{I}(\eta) \leftarrow \{i \in [N] : \tilde{r}_i \ge 0\}$.
4: $S(\eta) \leftarrow$ StaticMNL$(\tilde{\boldsymbol{r}}, \boldsymbol{v}, \mathcal{I}(\eta) \cap \mathcal{S}_K)$ by Algorithm 6.
5: $H(\eta) \leftarrow H(\eta; S(\eta), \boldsymbol{v})$.
6: **Output:** $(S(\eta), H(\eta))$.

---

**Step 2: Outer Subproblem (Golden Section Search).** With the inner maximization efficiently handled, our task reduces to maximizing the univariate function

$$H^{\star}(\eta) := \max_{S \in \mathcal{I}(\eta) \cap \mathcal{S}_K} H(\eta; S, \boldsymbol{v}).$$

In contrast to the KL case, the objective $H^{\star}(\eta)$ is in general not monotone in $\eta$, so bisection search is not applicable. Empirically, however, we observe that $H^{\star}(\eta)$ is unimodal in the vast majority of randomly generated instances (see Appendix D.2). Motivated by this observation, we employ the Golden Section Search (GSS) method over the interval $[0, \frac{\sqrt{1+\delta}}{\sqrt{1+\delta}-1}]$ as a fast practical heuristic. GSS is a derivative-free method that iteratively narrows the search bracket by evaluating the function $\mathcal{H}(\eta)$ at specific test points, where each evaluation is carried out via the EVAL-H subroutine (Algorithm 4).

By combining the exact solution of the inner static MNL problem with the $\epsilon$-accuracy guarantee of the outer Golden Section Search, our overall procedure (Algorithm 5) returns an $\epsilon$-optimal robust assortment.

**Computational Complexity.** Since GSS shrinks the search interval by a constant factor $\varphi \approx 0.618$ per iteration, it requires $O(\log(1/\epsilon))$ evaluations to reach precision $\epsilon$. Each evaluation calls StaticMNL, which can be implemented in $O(N \log N)$ time. Therefore, the total complexity is $O(N \log N \cdot \log(1/\epsilon))$. We empirically verify this scaling in Appendix D.1.

## 4. Data-Driven DRAO with Contextual MNL

In this section, we extend the distributionally robust assortment optimization framework to the *contextual* MNL setting. In practice, the true underlying parameter $\boldsymbol{\theta}^{\star}$ is unknown, meaning the attraction values $\boldsymbol{v} = \exp(\mathbf{X}^{\top}\boldsymbol{\theta}^{\star})$ and the corresponding choice probabilities $p(\cdot|S, \boldsymbol{\theta}^{\star})$ are inaccessible. Our offline goal is to construct an assortment $\hat{S}_D \in \mathcal{S}_K$ whose robust revenue under the true model is close to the optimal robust value.

Our methodology proceeds in three stages. First, we employ a statistical estimator for $\boldsymbol{\theta}^{\star}$ to construct a pessimistic

**Algorithm 5** $\chi^2$-Robust MNL

> **Input:** Error tolerance $\epsilon$.
> **Initialize:** $\varphi \leftarrow \frac{\sqrt{5}-1}{2}$, $\eta_L \leftarrow 0$, $\eta_R \leftarrow \frac{\sqrt{1+\delta}}{\sqrt{1+\delta}-1}$.
> $\eta_1 \leftarrow \eta_R - \varphi(\eta_R - \eta_L)$, $\eta_2 \leftarrow \eta_L + \varphi(\eta_R - \eta_L)$.
> $(S_1, H_1) \leftarrow \text{EVAL-H}(\eta_1)$, $(S_2, H_2) \leftarrow \text{EVAL-H}(\eta_2)$.
> **while** $(\eta_R - \eta_L) > \epsilon$ **do**
>   **if** $H_1 < H_2$ **then**
>     $\eta_L \leftarrow \eta_1, \eta_1 \leftarrow \eta_2, H_1 \leftarrow H_2$.
>     $\eta_2 \leftarrow \eta_L + \varphi(\eta_R - \eta_L)$.
>     $(S_2, H_2) \leftarrow \text{EVAL-H}(\eta_2)$.
>   **else**
>     $\eta_R \leftarrow \eta_2, \eta_2 \leftarrow \eta_1, H_2 \leftarrow H_1$.
>     $\eta_1 \leftarrow \eta_R - \varphi(\eta_R - \eta_L)$.
>     $(S_1, H_1) \leftarrow \text{EVAL-H}(\eta_1)$.
>   **end if**
> **end while**
> $\eta \leftarrow (\eta_R + \eta_L)/2$.
> $(S, H) \leftarrow \text{EVAL-H}(\eta_1)$
> **Return:** $S_{\chi^2}^\epsilon \leftarrow S$

estimate of the attraction values, denoted as $\boldsymbol{v}^{\text{LCB}}$. Then, we plug $v^{\text{LCB}}$ into the robust optimization procedures developed in Section 3 to obtain $\hat{S}_D$. Finally, we provide a theoretical analysis of the high-probability sub-optimality bounds.

### 4.1. Pessimistic Choice Model

Following established literature (Han et al., 2025a; Erginbas et al., 2025; Perivier & Goyal, 2022; Oh & Iyengar, 2021; Agrawal et al., 2023; Miao & Chao, 2021), we utilize a $\lambda$-regularized Maximum Likelihood Estimator (MLE) to estimate the unknown parameter $\boldsymbol{\theta}^\star$. Given a historical dataset $\{c_t, \boldsymbol{X}_t, S_t\}_{t=1}^T$, define the $\lambda$-regularized log-likelihood function as

$$\ell_T^\lambda(\boldsymbol{\theta}) = \sum_{t=1}^T \sum_{j \in (S_t)_+} \mathbf{1}\{c_t = j\} \log p(i|S_t, \boldsymbol{\theta}) - \frac{\lambda}{2}\|\boldsymbol{\theta}\|_2^2.$$

Then, the $\lambda$-regularized MLE is given by

$$\hat{\boldsymbol{\theta}}_T^\lambda := \operatorname*{argmax}_{\boldsymbol{\theta} \in \mathbb{R}^d} \ell_T^\lambda(\boldsymbol{\theta}), \tag{11}$$

For confidence-region construction, we define the Hessian matrix $\boldsymbol{H}_T^\lambda(\boldsymbol{\theta}) := \nabla_{\boldsymbol{\theta}}^2 \ell_T^\lambda(\boldsymbol{\theta})$, which is positive semidefinite under the MNL model.

To control the estimation error of $\boldsymbol{\theta}^\star$, we adopt the following standard assumptions. Under these conditions, we restate the confidence bound established by Han et al. (2025a), which serves as the basis for our contextual robust analysis.

**Assumption 4.1** (Independence). Conditioned on the assortments and contexts $\{\boldsymbol{X}_t, S_t\}_{t=1}^T$, the observed choices $\{c_t\}_{t=1}^T$ are mutually independent.

**Assumption 4.2** (Burn-in Condition). The collected dataset $\{c_t, \boldsymbol{X}_t, S_t\}_{t=1}^T$ satisfies the burn-in condition

$$\max_{t \le T, j \in S_t} \|\boldsymbol{x}_{tj}\|_{\boldsymbol{H}_T^\lambda(\boldsymbol{\theta}^\star)^{-1}} \le \frac{1}{144\sqrt{d\log(N/\zeta)}} \wedge \frac{1}{24\sqrt{\lambda}W}.$$

**Theorem 4.3** (Theorem 1 in Han et al. (2025a)). *Under Assumptions 2.1, 4.1 and 4.2, with probability at least $1 - \zeta$, for any $\boldsymbol{x} \in \mathbb{R}^d$ satisfying $\|\boldsymbol{x}\| \le 1$, it holds that*

$$|\boldsymbol{x}^\top(\hat{\boldsymbol{\theta}}_T^\lambda - \boldsymbol{\theta}^\star)| \le 16\|\boldsymbol{x}\|_{\boldsymbol{H}_T^\lambda(\boldsymbol{\theta}^\star)^{-1}}\left(\sqrt{\log(N/\zeta)} + \sqrt{\lambda}W\right).$$

*Remark* 4.4. Assumption 4.2 is sufficient but not necessary for Theorem 4.3. For instance, if the assortments $\{S_t\}_{t=1}^T$ are sampled i.i.d. from a fixed exploration policy $\pi$, a standard model for randomized data collection in retail and recommendation systems, then the burn-in condition is satisfied asymptotically as $T \to \infty$. More broadly, our LCB construction in (12) and the sub-optimality analyses in Section 4.2 only require a confidence bound of the form $|\boldsymbol{x}^\top(\hat{\boldsymbol{\theta}}_T^\lambda - \boldsymbol{\theta}^\star)| \le \beta_T\|\boldsymbol{x}\|_{\boldsymbol{H}_T^\lambda(\boldsymbol{\theta}^\star)^{-1}}$ for some data-dependent parameter $\beta_T$. Any such bound can be substituted for Theorem 4.3 to obtain an analogous sub-optimality guarantee. We adopt Han et al. (2025a) because it currently provides the tightest such bound for offline contextual MNL.

**Lower Confidence Bound (LCB) Attractions.** Based on Theorem 4.3, we construct a pessimistic estimate of the item attraction value. Specifically, for any item $i$ with feature vector $\boldsymbol{x}_i$, we define the LCB attraction $v_i^{\text{LCB}}$ as:

$$v_i^{\text{LCB}} = \begin{cases} \exp(\boldsymbol{x}_i^\top \hat{\boldsymbol{\theta}}_T^\lambda - \xi_i) & \text{if } i \in \cup_{t=1}^T S_t \setminus \{0\}, \\ 0 & \text{otherwise,} \end{cases} \tag{12}$$

where $\xi_i := 16\|\boldsymbol{x}_i\|_{\boldsymbol{H}_T^\lambda(\boldsymbol{\theta}^\star)^{-1}}\left(\sqrt{\log(N/\zeta)} + \sqrt{\lambda}W\right)$. Given an assortment $S$, the pessimistic choice probabilities induced by $\boldsymbol{v}^{\text{LCB}}$ are

$$\hat{p}(i|S) := \frac{v_i^{\text{LCB}}}{1 + \sum_{j \in S} v_j^{\text{LCB}}}, \quad i \in S_+.$$

**Data-Driven Robust Assortment Selection.** After constructing the pessimistic attraction $\boldsymbol{v}^{\text{LCB}}$, we select the robust assortment by solving

$$\hat{S}_D := \operatorname*{argmax}_{S \in \mathcal{S}_K} R_D^\delta(S|\boldsymbol{v}^{\text{LCB}}). \tag{13}$$

where $R_D^\delta(\cdot)$ is the worst-case revenue under $f$-divergence $D \in \{\text{TV}, \text{KL}, \chi^2\}$. This optimization can be efficiently solved using the polynomial-time algorithms developed in Section 3. Specifically, we apply Algorithm 1 for the TV ambiguity set. For the KL and $\chi^2$ ambiguity sets, we run Algorithm 2 and Algorithm 5 respectively, and compute an $\varepsilon$-accurate solution for any prescribed numerical tolerance $\varepsilon > 0$.

## 4.2. Sub-optimality Gap Guarantees

In this section, we characterize the performance of the robust assortment $\hat{S}_D$ by establishing high-probability upper bounds on the sub-optimality gap. For the purpose of our theoretical analysis, $\hat{S}_D$ denotes the exact optimizer of (13). For clarity, we use $A \lesssim B$ to denote that $A \leq c \cdot B$ for some universal constant $c$, omitting logarithmic factors $\log(N/\zeta)$ and the regularization parameter $\sqrt{\lambda}W$. The complete expressions with all constants are provided in Appendix C.

**Theorem 4.5** (TV Sub-optimality). *Under Assumptions 2.1, 4.1 and 4.2, with probability at least $1 - \zeta$,*

$$\text{SubOpt}_{\text{TV}}^{\delta}(\hat{S}_{\text{TV}}|\boldsymbol{\theta}^{\star})$$
$$\lesssim \sum_{i \in S_{\text{TV}}^{\star}} p(i|S_{\text{TV}}^{\star}, \boldsymbol{\theta}^{\star})\|\boldsymbol{x}_i\|_{\boldsymbol{H}_T^{\lambda}(\boldsymbol{\theta}^{\star})^{-1}} + \max_{i \in S_{\text{TV}}^{\star}} \|\boldsymbol{x}_i\|_{\boldsymbol{H}_T^{\lambda}(\boldsymbol{\theta}^{\star})^{-1}}^2.$$

**Theorem 4.6** (KL Sub-optimality). *Under Assumptions 2.1, 4.1, 4.2 and the small-gap condition*

$$\max_{i \in [N]} 32\|\boldsymbol{x}_i\|_{\boldsymbol{H}_T^{\lambda}(\boldsymbol{\theta}^{\star})^{-1}} \left( \sqrt{\log(N/\zeta)} + \sqrt{\lambda}W \right) \leq 1,$$

*with probability at least $1 - \zeta$,*

$$\text{SubOpt}_{\text{KL}}^{\delta}(\hat{S}_{\text{KL}}|\boldsymbol{\theta}^{\star})$$
$$\lesssim \min\left\{ \frac{1}{\log(3/2)}, \frac{1}{\delta} \right\} \sum_{i \in S_{\text{KL}}^{\star}} p(i|S_{\text{KL}}^{\star}, \boldsymbol{\theta}^{\star})\|\boldsymbol{x}_i\|_{\boldsymbol{H}_T^{\lambda}(\boldsymbol{\theta}^{\star})^{-1}}.$$

**Theorem 4.7** ($\chi^2$ Sub-optimality). *Under Assumptions 2.1, 4.1 and 4.2, with probability at least $1 - \zeta$,*

$$\text{SubOpt}_{\chi^2}^{\delta}(\hat{S}_{\chi^2}|\boldsymbol{\theta}^{\star}) \lesssim \frac{1+\delta}{\sqrt{1+\delta}-1} \cdot$$
$$\sqrt{\sum_{i \in S_{\chi^2}^{\star}} p(i|S_{\chi^2}^{\star}, \boldsymbol{\theta}^{\star})\|\boldsymbol{x}_i\|_{\boldsymbol{H}_T^{\lambda}(\boldsymbol{\theta}^{\star})^{-1}} + \max_{i \in S_{\chi^2}^{\star}} \|\boldsymbol{x}_i\|_{\boldsymbol{H}_T^{\lambda}(\boldsymbol{\theta}^{\star})^{-1}}^2}$$

**Proof Sketch.** All three bounds in Theorems 4.5–4.7 follow from a unified two-term decomposition:

$$\text{SubOpt}_D^{\delta}(\hat{S}_D|\boldsymbol{\theta}^{\star}) \leq \underbrace{R_D^{\delta}(S_D^{\star}|\boldsymbol{v}) - R_D^{\delta}(S_D^{\star}|\boldsymbol{v}^{\text{LCB}})}_{\text{(I) sensitivity}}$$
$$+ \underbrace{R_D^{\delta}(\hat{S}_D|\boldsymbol{v}^{\text{LCB}}) - R_D^{\delta}(\hat{S}_D|\boldsymbol{v})}_{\text{(II) monotonicity}}.$$

Term (I) measures the sensitivity of the robust revenue at the fixed assortment $S_D^{\star}$ to the perturbation $\boldsymbol{v} \to \boldsymbol{v}^{\text{LCB}}$. Our dual representation in Section 3 strips off the robust layer and reduces this to a non-robust choice-probability gap, which is then bounded by the utility gap from Theorem 4.3 via Lemma C.1. Term (II) is non-positive due to a monotonicity property of the robust revenue at the LCB optimizer $\hat{S}_D$ on the high-probability event $\boldsymbol{v}^{\text{LCB}} \leq \boldsymbol{v}$. Divergence-specific proofs are deferred to Appendix C.

**Interpretation of Bounds.** The bounds in Theorems 4.5–4.7 are controlled by self-normalized quantities of the form $\|\boldsymbol{x}_i\|_{\boldsymbol{H}_T^{\lambda}(\boldsymbol{\theta}^{\star})^{-1}}$ and their squared counterparts. Hence, the guarantees shrink as the information matrix $\boldsymbol{H}_T^{\lambda}(\boldsymbol{\theta}^{\star})$ becomes better conditioned, equivalently, as $\|\boldsymbol{x}_i\|_{\boldsymbol{H}_T^{\lambda}(\boldsymbol{\theta}^{\star})^{-1}}$ decreases for items appearing in the optimal robust assortment.

Moreover, suppose one can relate the information matrix $\boldsymbol{H}_T^{\lambda}(\boldsymbol{\theta}^{\star})$ to the empirical design matrix $\boldsymbol{V}_T := \sum_{t=1}^{T} \sum_{i \in S_t} \boldsymbol{x}_{ti}\boldsymbol{x}_{ti}^{\top}$ via $\boldsymbol{H}_T^{\lambda}(\boldsymbol{\theta}^{\star}) \succeq \kappa \boldsymbol{V}_T$ for some $\kappa > 0$, which implies $\|\boldsymbol{x}_i\|_{\boldsymbol{H}_T^{\lambda}(\boldsymbol{\theta}^{\star})^{-1}} \leq \kappa^{-1/2}\|\boldsymbol{x}_i\|_{\boldsymbol{V}_T^{-1}}$. Under this comparison, our bounds implies that sub-optimality vanishes whenever $\max_{i \in S_D^{\star}} \|\boldsymbol{x}_i\|_{\boldsymbol{V}_T^{-1}} \to 0$. This provides a feature-based, *localized* notion of offline coverage that depends only on items in the optimal robust assortment. This requirement is significantly weaker than prior conditions that demand global, *assortment-wise* exploration of the action space (e.g., Dong et al., 2023).

Finally, retaining the choice-probability weights $p(i|S_D^{\star}, \boldsymbol{\theta}^{\star})$ avoids the looseness associated with worst-case relaxations. As discussed in Han et al. (2025a), replacing $\boldsymbol{H}_T^{\lambda}(\boldsymbol{\theta}^{\star})$ by $\boldsymbol{V}_T$ can introduce multiplicative constants that may scale exponentially with the maximum utility. Keeping probability-weighted terms helps mitigate such conservative dependence, in line with the sharper analyses of Han et al. (2025a).

**Numerical tolerance.** While our theoretical results analyze the exact optimizer $\hat{S}_D$, we note that the computational accuracy varies by ambiguity set. Algorithm 1 for the TV case is exact and incurs no numerical error. In contrast, the optimizations for the KL and $\chi^2$ ambiguity sets rely on the one-dimensional search techniques in Algorithms 2 and 5 to achieve $\varepsilon$-optimality. Consequently, the sub-optimality gap for the computed assortments includes an additional additive term of $\varepsilon$. In practice, $\varepsilon$ can be chosen to be significantly smaller than the statistical estimation error (e.g., $\varepsilon = 10^{-6}$), so its effect can be negligible.

**Prefactor Scaling in $\delta$.** The three $f$-divergences yield upper bounds with distinct prefactor dependence on the ambiguity radius $\delta$. For TV, the bound scales linearly with the estimation error and is independent of $\delta$, owing to the piecewise-linear structure of the TV dual over a finite set of reward breakpoints. For KL, the $\delta$-dependence is captured by $\min\left\{ \frac{1}{\log(3/2)}, \frac{1}{\delta} \right\}$: for $\delta \leq \log(3/2)$ the prefactor remains constant, while for larger $\delta$ it follows a $1/\delta$ scaling, so the bound tightens as the ambiguity set grows. For $\chi^2$, the prefactor $\frac{1+\delta}{\sqrt{1+\delta}-1}$ behaves like $2/\delta$ for small $\delta$. We emphasize that the larger KL and $\chi^2$ prefactors are features of our *upper bounds*: they originate from bounding the dual perturbation uniformly over a continuous dual interval, in contrast to the finite-breakpoint TV dual. Without a match-

ing lower bound, these prefactors alone do not imply that the KL- or $\chi^2$-robust policies are inherently less stable than their TV counterpart.

## Acknowledgements

This work is generously supported by ONR-13983263 and 2027 New York University Center for Global Economy and Business grant.

## Impact Statement

This paper presents work whose goal is to advance the field of Machine Learning. There are many potential societal consequences of our work, none which we feel must be specifically highlighted here.

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

# A. Proof of Strong Duality

## A.1. Proof of Results under TV Ambiguity Set

***Proof of Lemma 3.1***. Given an assortment $S \in \mathcal{S}_K$ and a nominal attraction value $\boldsymbol{v}$, the derivation of Equation (5) follows from Lemma 5 in (Panaganti et al., 2022). Specifically, the robust revenue can be expressed as

$$R_{\mathrm{TV}}^{\delta}(S|\boldsymbol{v}) = \min_{q \in \Delta(S_+), D_{\mathrm{TV}}(q,\, p(\cdot|S, \boldsymbol{v})) \leq \delta} \mathbb{E}_{i \sim q}[r_i]$$

$$= -\min_{\alpha \in \mathbb{R}} \left\{ \mathbb{E}_{i \sim p(\cdot|S, \boldsymbol{v})} \left[ (\alpha - r_i)_+ \right] + \delta \left( \alpha - \min_{i \in S_+} r_i \right)_+ - \alpha \right\}$$

$$= -\min_{\alpha \in \mathbb{R}} \left\{ \mathbb{E}_{i \sim p(\cdot|S, \boldsymbol{v})} \left[ (\alpha - r_i)_+ \right] + \delta \alpha_+ - \alpha \right\},$$

where the last line holds because $r_0 = 0$ and $r_i \in [0, 1]$ implies $\min_{i \in S_+} r_i = 0$.

Let $\phi(\alpha) = \mathbb{E}_{i \sim p(\cdot|S, \boldsymbol{v})} \left[ (\alpha - r_i)_+ \right] + \delta \alpha_+ - \alpha$ denote the objective function inside the infimum. Note that $\phi(\alpha)$ is convex in $\alpha$. For $\alpha \leq 0$, we have $\phi(\alpha) = -\alpha \geq 0$ and $h(0) = 0$, which implies that $\min_{\alpha \in (-\infty, 0]} = 0$ is attained at $\alpha = 0$. On the other hand, $h(1/\delta) = \mathbb{E}_{i \sim p(\cdot|S, \boldsymbol{v})} \left[ (1/\delta - r_i) \right] + 1 - 1/\delta = 1 - \mathbb{E}_{i \sim p(\cdot|S, \boldsymbol{v})}[r_i] \geq 0$. (WLOG, we assume $\delta \in [0, 1]$ is a small positive number since $D_{\mathrm{TV}}(q, p) \leq 1$ for all distributions.) Thus, it suffices to restrict the above optimization problem to $\alpha \in [0, 1/\delta]$, i.e.,

$$R_{\mathrm{TV}}^{\delta}(S|\boldsymbol{v}) = -\min_{\alpha \in [0, 1/\delta]} \left\{ \mathbb{E}_{i \sim p(\cdot|S, \boldsymbol{v})} \left[ (\alpha - r_i)_+ \right] - (1 - \delta)\alpha \right\}.$$

To prove Equation (6), we reformulate the minimization problem as the maximization of a concave function. Define $\Phi(\alpha; S, \boldsymbol{v}) = -\phi(\alpha)$, specifically:

$$\Phi(\alpha; S, \boldsymbol{v}) := -\mathbb{E}_{i \sim p(\cdot|S, \boldsymbol{v})} \left[ (\alpha - r_i)_+ \right] + (1 - \delta)\alpha$$

$$= \mathbb{E}_{i \sim p(\cdot|S, \boldsymbol{v})} \left[ \min\{\alpha, r_i\} \right] - \delta\alpha$$

where the second line holds due to $-(\alpha - r_i)_+ = \min\{\alpha, r_i\} - \alpha$. Then, we analyze the value of $R_{\mathrm{TV}}^{\delta}(S|\boldsymbol{v}) = -\min_{\alpha \in \mathbb{R}} -\Phi(\alpha; S, \boldsymbol{v}) = \max_{\alpha \in \mathbb{R}} \Phi(\alpha; S, \boldsymbol{v})$. The function $\Phi(\alpha; S, \boldsymbol{v})$ is concave, its subgradient $\Phi'(\alpha; S, \boldsymbol{v})$ is non-increasing as $\alpha$ increases, where

$$\Phi'(\alpha; S, \boldsymbol{v}) = -\mathbb{P}_{i \sim p(\cdot|S, \boldsymbol{v})}(r_i < \alpha) + 1 - \delta = \mathbb{P}_{i \sim p(\cdot|S, \boldsymbol{v})}(r_i \geq \alpha) - \delta$$

The global maximum occurs at the point $\alpha^*$ where the subgradient contains zero. Setting $\Phi'(\alpha^*; S, \boldsymbol{v}) = 0$, we find that optimal $\alpha^*$ corresponds to the $(1 - \delta)$-quantile of the reward distribution:

$$\alpha^* = \min \left\{ \alpha : \mathbb{P}_{i \sim p(\cdot|S, \boldsymbol{v})}(r_i \geq \alpha) \leq \delta \right\} \in [0, \max_{i \in S_+} r_i] \subseteq [0, 1/\delta].$$

Since the support of the reward distribution is discrete, we may not find $\alpha^*$ such that $\Phi'(\alpha^*; S, \boldsymbol{v}) = 0$, but the maximizer can still be found by examining these breakpoints. We sort the rewards within assortment $S$: $r_1 \leq r_2 \leq \cdots \leq r_{|S|}$. For any $\alpha \in (r_j, r_{j+1})$, the subgradient is constant:

$$\Phi'(\alpha; S, \boldsymbol{v}) = \mathbb{P}_{i \sim p(\cdot|S, \boldsymbol{v})}(r \geq r_j) - \delta = \sum_{i \in S: i \geq j} p(i|S, \boldsymbol{v}) - \delta.$$

We define the critical index $k$ as the smallest index satisfying the probability threshold:

$$k = \min \left\{ j \in S : \sum_{i \in S: i \geq j} p(i|S, \boldsymbol{v}) \leq \delta \right\}.$$

Hence, the optimization reduces to evaluating the function at the discrete reward points:

$$R_{\mathrm{TV}}^{\delta}(S|\boldsymbol{v}) = \max_{\alpha \in [0, 1/\delta]} \Phi(\alpha; S, \boldsymbol{v}) = \max_{1 \leq i \leq |S|} \{\Phi(r_i; S, \boldsymbol{v})\} = \Phi(r_k; S, \boldsymbol{v}), \tag{14}$$

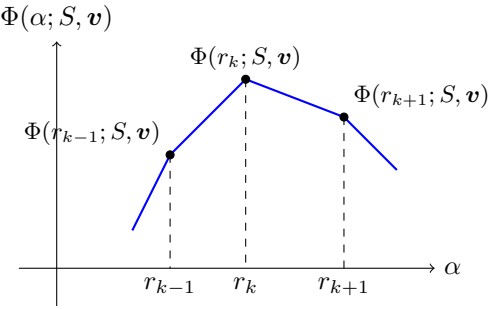

*Figure 1.* Piecewise linear function $\Phi(\alpha; S, \boldsymbol{v})$.

Consequently, $\Phi(\alpha; S, \boldsymbol{v})$ is a piecewise linear function with a unique maximum at $r_k$. This characterization derives from the following threshold conditions:

$$\sum_{i \in S: i \geq k} p(i|S, \boldsymbol{v}) \leq \delta, \quad \text{whereas} \quad \sum_{i \in S: i \geq k-1} p(i|S, \boldsymbol{v}) > \delta.$$

These inequalities yield the derivative behavior of $\Phi$:

$$\Phi'(\alpha; S, \boldsymbol{v}) = \begin{cases} \sum_{i \in S: i \geq k-1} p(i|S, \boldsymbol{v}) - \delta > 0, & \alpha \in (r_{k-1}, r_k), \\ \sum_{i \in S: i \geq k} p(i|S, \boldsymbol{v}) - \delta \leq 0, & \alpha \in (r_k, r_{k+1}), \end{cases}$$

confirming that $r_k$ is indeed the maximizer.

where the optimal value is explicitly given by:

$$\Phi(r_k; S, \boldsymbol{v}) = - \sum_{i \in S: i < k} p(i|S, \boldsymbol{v}) \cdot (r_k - r_i) + (1 - \delta) r_k.$$

Equivalently, we can also write $R_{\text{TV}}^{\delta}(S|\boldsymbol{v}) = \max_{i \in [N]} \{\Phi(r_i; S, \boldsymbol{v})\}$.

$\square$

## A.2. Proof of Results under KL Ambiguity Set

***Proof of Lemma 3.3.*** The first equality in Equation (7) follows directly from Theorem 1 in (Hu & Hong, 2013). Note that the objective function inside the infimum $g(\rho) = \rho \log \left(\mathbb{E}_{i \sim p(\cdot|S, \boldsymbol{v})}[\exp(-r_i/\rho)]\right) + \rho \delta$ is convex in $\rho$. We now derive a bound for the optimal $\rho$. Since $r_i \leq 1$ for all $i$, for any $\rho \geq 1/\delta$, we have

$$g(\rho) = \rho \log \left(\mathbb{E}_{i \sim p(\cdot|S, \boldsymbol{v})}[\exp(-r_i/\rho)]\right) + \rho \delta \geq \rho \log \left(\mathbb{E}_{i \sim p(\cdot|S, \boldsymbol{v})}[\exp(-1/\rho)]\right) + \rho \delta = \rho \delta - 1 \geq 0.$$

In addition, we observe that there exist $\rho \in (0, 1/\delta)$ satisfying $g(\rho) < 0$. Thus, it suffices to optimize $\rho$ over $[0, 1/\delta]$, which implies that

$$\begin{aligned} R_{\text{KL}}(S|\boldsymbol{v}) &= - \min_{0 \leq \rho \leq 1/\delta} \left\{\rho \log \left(\mathbb{E}_{i \sim p(\cdot|S, \boldsymbol{v})}[\exp(-r_i/\rho)]\right) + \rho \delta\right\} \\ &= \max_{0 \leq \rho \leq 1/\delta} \left\{-\rho \log \left(\mathbb{E}_{i \sim p(\cdot|S, \boldsymbol{v})}[\exp(-r_i/\rho)]\right) - \rho \delta\right\} \end{aligned}$$

$\square$

## A.3. Proof of Results under Chi-square Ambiguity Set

***Proof of Lemma 3.4.*** By applying Lemma 9 in (Xu et al., 2023), we directly establish the first equality in Equation (9). Define the dual objective function as $h(\eta) = \sqrt{(1 + \delta)\mathbb{E}_{i \sim p(\cdot|S, \boldsymbol{v})}[(\eta - r_i)^2]} - \eta$. Note that $h$ is convex in dual variable $\eta$, and $h(\eta) \geq 0$ when $\eta \leq 0$. In addition, since $r_i \in [0, 1]$ for all $i \in [N]$, we have

$$h(\frac{\sqrt{1+\delta}}{\sqrt{1+\delta}-1}) \geq \sqrt{(1+\delta)\mathbb{E}_{i \sim p(\cdot|S, \boldsymbol{v})}\left[\left(\frac{\sqrt{1+\delta}}{\sqrt{1+\delta}-1} - 1\right)^2\right]} - \frac{\sqrt{1+\delta}}{\sqrt{1+\delta}-1} = 0.$$

Hence, the optimal dual variable must lie within the interval $\eta \in [0, \frac{\sqrt{1+\delta}}{\sqrt{1+\delta}-1}]$. It follows that

$$R_{\chi^2}^{\delta}(S|\boldsymbol{v}) = - \min_{0 \leq \eta \leq \frac{\sqrt{1+\delta}}{\sqrt{1+\delta}-1}} \left\{\sqrt{(1 + \delta)\mathbb{E}_{i \sim p(\cdot|S, \boldsymbol{v})}[(\eta - r_i)^2]} - \eta\right\}.$$

$\square$

# B. Results in Algorithm Design

## B.1. `StaticMNL` Algorithm

In this section, we present `StaticMNL` introduced by Rusmevichientong et al. (2010).

---

**Algorithm 6** StaticMNL((Rusmevichientong et al., 2010))

---

1: **Input:** Revenue $r$, attraction value $v$, candidate set $\mathcal{S}_K$.
2: **Define intersection points:**

- For each $j \in [N]$, define $I(0, j) := r_j$.

- For each $i, j \in [N]$ with $i \neq j$, define $I(i, j) := \frac{v_i r_i - v_j r_j}{v_i - v_j}$.

3: Let $L := \frac{N(N+1)}{2}$. Construct the set of index pairs

$$\mathcal{P} := \{(0, j) : j \in [N]\} \cup \{(i, j) : i, j \in [N], i < j\},$$

and sort $\mathcal{P}$ into $(i_1, j_1), \ldots, (i_L, j_L)$ such that

$$-\infty \equiv I(i_0, j_0) < I(i_1, j_1) \leq \cdots \leq I(i_L, j_L) < I(i_{L+1}, j_{L+1}) \equiv +\infty.$$

4: **Initialize ordering:** Let $\boldsymbol{\sigma}^0 = (\sigma_1^0, \ldots, \sigma_N^0)$ be a permutation of $[N]$ such that $v_{\sigma_1^0} \geq v_{\sigma_2^0} \geq \cdots \geq v_{\sigma_N^0}$.
5: **Initialize:** $B_0 \leftarrow \emptyset$, $G_0 \leftarrow \{\sigma_1^0, \ldots, \sigma_K^0\}$, $A_0 \leftarrow G_0 \setminus B_0$.
6: **for** $t = 1, 2, \ldots, L$ **do**
7:    **if** $i_t \neq 0$ **then**
8:       Let $\boldsymbol{\sigma}^t$ be the permutation obtained from $\boldsymbol{\sigma}^{t-1}$ by swapping the positions of items $i_t$ and $j_t$.
9:       Set $B_t \leftarrow B_{t-1}$.
10:   **else**
11:       Set $\boldsymbol{\sigma}^t \leftarrow \boldsymbol{\sigma}^{t-1}$.
12:       Set $B_t \leftarrow B_{t-1} \cup \{j_t\}$.
13:   **end if**
14:   Set $G_t \leftarrow \{\sigma_1^t, \ldots, \sigma_K^t\}$ and $A_t \leftarrow G_t \setminus B_t$.
15: **end for**
16: **Select best candidate:** $\ell^\star \in \arg\max_{\ell \in \{0, 1, \ldots, L\}} R(A_\ell \mid \boldsymbol{v})$.
17: **Return:** $S^\star \leftarrow A_{\ell^\star}$.

---

## B.2. Auxiliary Lemmas under KL Ambiguity Set

**Lemma B.1.** *Let $g_i(t, \rho) := v_i \left( \exp((t - r_i)/\rho) - \exp(-\delta) \right)$, and define $\mathcal{G}(t) := \min_{S \subseteq \mathcal{N}(t) \cap \mathcal{S}_K, \rho \in [0, 1/\delta]} \sum_{i \in S_+} g_i(t, \rho)$ with $\mathcal{N}(t) := \{i \in [N] : t \leq r_i\}$. Then, the optimal KL-robust revenue $R_{\mathrm{KL}}^\star$ is equivalent to:*

$$R_{\mathrm{KL}}^\star = \max_{0 \leq t \leq 1} \{\mathcal{G}(t) \leq 0\}.$$

*Proof of Lemma B.1.* By the dual reformulation established in Lemma 3.3, the optimal KL-robust revenue can be written as

$$R_{\mathrm{KL}}^\star := \max_{S \in \mathcal{S}_K} R_{\mathrm{KL}}^\delta(S|\boldsymbol{v}) = \max_{S \in \mathcal{S}_K} \max_{0 \leq \rho \leq 1/\delta} \left\{ -\rho \log \left( \mathbb{E}_{i \sim p(\cdot | S, \boldsymbol{v})}[\exp(-r_i/\rho)] \right) - \rho\delta \right\}$$

$$= \max_{S \in \mathcal{S}_K, 0 \leq \rho \leq 1/\delta} \left\{ -\rho \log \left( \frac{\sum_{i \in S_+} v_i \exp(-r_i/\rho)}{\sum_{j \in S_+} v_j} \right) - \rho\delta \right\}.$$

Since the revenues are normalized to $r_i \in [0, 1]$, we restrict the threshold variable to $t \in [0, 1]$ and rewrite the maximization

as following,

$$R_{\mathrm{KL}}^\star = \max_{0 \le t \le 1} \left\{ \exists S \in \mathcal{S}_K, \rho \in [0, 1/\delta], -\rho \log \left( \frac{\sum_{i \in S_+} v_i \exp(-r_i/\rho)}{\sum_{j \in S_+} v_j} \right) - \rho \delta \ge t \right\}$$

$$= \max_{0 \le t \le 1} \left\{ \exists S \in \mathcal{S}_K, \rho \in [0, 1/\delta], \sum_{i \in S_+} v_i \left( \exp((t - r_i)/\rho) - \exp(-\delta) \right) \le 0 \right\}$$

$$= \max_{0 \le t \le 1} \left\{ \min_{S \in \mathcal{S}_K, \rho \in [0, 1/\delta]} \sum_{i \in S_+} g_i(t, \rho) \le 0 \right\}.$$

Note that if $r_i < t$ then $(t - r_i)/\rho > 0$ and $\exp((t - r_i)/\rho) > \exp(-\delta)$ for any $\rho > 0$, implying $g_i(t, \rho) > 0$. Hence, including any item with $r_i < t$ can only increase $\sum_{i \in S_+} g_i(t, \rho)$, and the minimum is attained by an assortment $S \subseteq \mathcal{N}(t) = \{i \in [N] : t \le r_i\}$. This yields the statement

$$R_{\mathrm{KL}}^\star = \max_{0 \le t \le 1} \left\{ \min_{S \in \mathcal{N}(t) \cap \mathcal{S}_K, \rho \in [0, 1/\delta]} \sum_{i \in S_+} g_i(t, \rho) \le 0 \right\} = \max_{0 \le t \le 1} \left\{ \mathcal{G}(t) \le 0 \right\}.$$

$\square$

**Lemma B.2.** *Fix $0 \le t \le 1$, the number of intersection points among the curves $\{g_i(t, \rho)\}_{\rho \ge 0, i \in \mathcal{N}(t)}$ is less than $\binom{|\mathcal{N}(t)|}{2}$. Furthermore, for any assortment set $S \subseteq [N]$, $\sum_{i \in S_+} g_i(t, \rho)$ is a quasi-convex function in $\rho$.*

***Proof of Lemma B.2.*** We first prove that the total number of intersection points is bounded by $\binom{|\mathcal{N}(t)|}{2}$. It suffices to show that any two distinct curve $g_i(t, \rho)$ and $g_j(t, \rho)$ intersect at no more than once in $(0, +\infty)$. Consider distinct items $i, j \in [N]$ such that $(v_i, r_i) \ne (v_j, r_j)$. WLOG, we assume that $v_i \ge v_j$ and then we analyze the following two different cases.

- Case 1: $v_i = v_j$. In this case, since $(v_i, r_i) \ne (v_j, r_j)$, it must hold that $r_i \ne r_j$. The condition $g_i(t, \rho) = g_j(t, \rho)$ simplifies to $\exp((t - r_i)/\rho) = \exp((t - r_j)/\rho)$, which implies $r_i = r_j$, a contradiction. Thus, the curves can only intersect at the limit $\rho \to 0$, representing at most one intersection point.

- Case 2: $v_i > v_j$. In this case, if two curves intersect, then we have

$$v_i \exp((t - r_i)/\rho) - v_j \exp((t - r_j)/\rho) = \exp(-\delta)(v_i - v_j).$$

The existence of such intersection point is equivalent to the existence of $\rho_{ij} \in [0, 1/\delta]$ such that

$$\varphi_{ij}(\rho) := v_i \exp((t - r_i)/\rho) - v_j \exp((t - r_j)/\rho) - \exp(-\delta)(v_i - v_j) = 0.$$

Note that $\lim_{\rho \to 0} \varphi_{ij}(\rho) = -\exp(-\delta)(v_i - v_j) < 0$ and $\lim_{\rho \to +\infty} \varphi_{ij}(\rho) = (1 - \exp(-\delta))(v_i - v_j) > 0$. By the Intermediate Value Theorem, there exists at least one root $\rho_{ij} \in (0, \infty)$. To prove uniqueness, we examine the derivative:

$$\frac{\partial}{\partial \rho} \varphi_{ij}(\rho) = \frac{1}{\rho^2} \cdot \left( v_i(r_i - t) \exp((t - r_i)/\rho) - v_j(r_j - t) \exp((t - r_j)/\rho) \right).$$

If $r_i = r_j$, then $\frac{\partial}{\partial \rho} \varphi_{ij}(\rho) = \frac{1}{\rho^2}(v_i - v_j)(r_i - t) \exp((t - r_i)/\rho) > 0$. Thus, $\varphi_{ij}(\rho)$ is strictly increasing and admits a unique root.

If $r_i \ne r_j$, the equation $\frac{\partial}{\partial \rho} \varphi_{ij}(\rho) = 0$ has at most one critical point $\rho^\star = \frac{r_i - r_j}{\log(v_i(r_i - t)) - \log(v_j(r_j - t))}$. Hence, $\varphi_{ij}(\rho)$ is monotone on the intervals $[0, \rho^\star]$ and $[\rho^\star, +\infty)$. Given the limits at the boundaries, it can cross zero at most once. Therefore, there is exactly one root.

In addition, the proof of the quasi-convexity of $\sum_{i \in S_+} g_i(t, \rho)$ follows the same approach as Lemma A.2 in Lu et al. (2026). $\square$

## B.3. Auxiliary Lemmas under Chi-square distance

**Lemma B.3.** *Fix* $0 \leq \eta \leq \frac{\sqrt{1+\delta}}{\sqrt{1+\delta}-1}$, *and define the transformed revenues* $\tilde{r}_i = 2\eta r_i - r_i^2$ *for all* $i \in [N]$ *with* $\tilde{r}_0 = 0$. *Let* $\mathcal{I}(\eta) := \{i \in [N] : \tilde{r}_i \geq 0\}$ *denote the set of items with non-negative transformed revenue. Then, the optimal* $\chi^2$-*robust revenue* $R_{\chi^2}^{\star}$ *is given by*

$$R_{\chi^2}^{\star} = \max_{0 \leq \eta \leq \frac{\sqrt{1+\delta}}{\sqrt{1+\delta}-1}} \left\{ \eta - \sqrt{(1+\delta)} \sqrt{\eta^2 - \max_{S \in \mathcal{I}(\eta) \cap \mathcal{S}_K} \mathbb{E}_{i \sim p(\cdot|S,\boldsymbol{v})}[\tilde{r}_i]} \right\}.$$

***Proof of Lemma B.3.*** By the strong duality established in Lemma 3.4, the optimal $\chi^2$-robust revenue can be expressed by interchanging the order of maximization:

$$R_{\chi^2}^{\star} := \max_{S \in \mathcal{S}_K} \max_{0 \leq \eta \leq \frac{\sqrt{1+\delta}}{\sqrt{1+\delta}-1}} H(\eta; S, \boldsymbol{v}) = \max_{0 \leq \eta \leq \frac{\sqrt{1+\delta}}{\sqrt{1+\delta}-1}} \max_{S \in \mathcal{S}_K} H(\eta; S, \boldsymbol{v})$$

$$= \max_{0 \leq \eta \leq \frac{\sqrt{1+\delta}}{\sqrt{1+\delta}-1}} \left\{ \eta - \sqrt{(1+\delta)} \sqrt{\min_{S \in \mathcal{S}_K} \mathbb{E}_{i \sim p(\cdot|S,\boldsymbol{v})}[(\eta - r_i)^2]} \right\}.$$

We now simplify the inner minimization problem. Expanding the quadratic term, we have

$$\min_{S \in \mathcal{S}_K} \mathbb{E}_{i \sim p(\cdot|S,\boldsymbol{v})}[(\eta - r_i)^2] = \min_{S \in \mathcal{S}_K} \sum_{i \in S_+} p(i|S,\boldsymbol{v})(\eta - r_i)^2 = \min_{S \in \mathcal{S}_K} \frac{\eta^2 + \sum_{i \in S} v_i(\eta - r_i)^2}{1 + \sum_{i \in S} v_i}$$

$$= \min_{S \in \mathcal{S}_K} \eta^2 - \frac{\sum_{i \in S} v_i(2\eta r_i - r_i^2)}{1 + \sum_{i \in S} v_i} = \eta^2 - \max_{S \in \mathcal{S}_K} \frac{\sum_{i \in S} v_i(2\eta r_i - r_i^2)}{1 + \sum_{i \in S} v_i}.$$

Substituting this into the objective yields

$$R_{\chi^2}^{\star} = \max_{0 \leq \eta \leq \frac{\sqrt{1+\delta}}{\sqrt{1+\delta}-1}} \left\{ \eta - \sqrt{(1+\delta)} \sqrt{\eta^2 - \max_{S \in \mathcal{S}_K} \frac{\sum_{i \in S} v_i \tilde{r}_i}{1 + \sum_{i \in S} v_i}} \right\} \tag{15}$$

We claim that any optimal assortment $\tilde{S} \in \operatorname{argmax}_{S \in \mathcal{S}_K} \mathbb{E}_{i \sim p(\cdot|S,\boldsymbol{v})}[\tilde{r}_i]$ must be a subset of $\mathcal{I}(\eta)$. Suppose, for the sake of contradiction, that there exists an item $j \in \tilde{S}$ such that $\tilde{r}_j < 0$. Then,

$$\frac{\sum_{i \in \tilde{S}} v_i \tilde{r}_i}{1 + \sum_{i \in \tilde{S}} v_i} = \frac{\sum_{i \in \tilde{S} \setminus \{j\}} v_i \tilde{r}_i + v_j \tilde{r}_j}{1 + \sum_{i \in \tilde{S}} v_i} < \frac{\sum_{i \in \tilde{S} \setminus \{j\}} v_i \tilde{r}_i}{1 + \sum_{i \in \tilde{S}} v_i} < \frac{\sum_{i \in \tilde{S} \setminus \{j\}} v_i \tilde{r}_i}{1 + \sum_{i \in \tilde{S} \setminus \{j\}} v_i}.$$

where the last inequality follows from the fact that the numerator $\sum_{i \in \tilde{S} \setminus \{j\}} v_i \tilde{r}_i$ is non-negative, otherwise the empty set would be superior to $\tilde{S}$. This strictly greater revenue for $\tilde{S} \setminus \{j\}$ contradicts the optimality of $\tilde{S}$. Thus, we may restrict the search space to $S \subseteq \mathcal{I}(\eta)$, which completes the proof. □

## C. Proof of Sub-Optimality Gap

We begin by establishing a general decomposition of the sub-optimality gap and defining the confidence intervals for the utility parameters that underpin our subsequent analysis.

Let $u_i := \boldsymbol{x}_i^\top \boldsymbol{\theta}^\star$, and $u_i^{\text{LCB}} := \boldsymbol{x}_i^\top \hat{\boldsymbol{\theta}}_T^\lambda - \xi_i$ with $\xi_i := 16\|\boldsymbol{x}_i\|_{\boldsymbol{H}_T^\lambda(\boldsymbol{\theta}^\star)^{-1}} \left( \sqrt{\log(N/\zeta)} + \sqrt{\lambda}W \right)$. According to Theorem 4.3, the utility gap is bounded by

$$|u_i - u_i^{\text{LCB}}| \leq 32\|\boldsymbol{x}_i\|_{\boldsymbol{H}_T^\lambda(\boldsymbol{\theta}^\star)^{-1}} \left( \sqrt{\log(N/\zeta)} + \sqrt{\lambda}W \right). \tag{16}$$

Recall that, by definition, we have

$$S_D^\star = \operatorname*{argmax}_{S \in \mathcal{S}_K} R_D^\delta(S|\boldsymbol{\theta}^\star) \quad \text{and} \quad \hat{S}_D = \operatorname*{argmax}_{S \in \mathcal{S}_K} R_D^\delta(S|\boldsymbol{v}^{\text{LCB}}).$$

Given $\boldsymbol{\theta}^\star$, let $\boldsymbol{v} = \exp(\boldsymbol{X}^\top \boldsymbol{\theta}^\star)$. By the definition of the sub-optimality gap, we have

$$
\begin{aligned}
\mathrm{SubOpt}_D^\delta(\hat{S}_D|\boldsymbol{\theta}^\star) &= R_D^\delta(S_D^\star|\boldsymbol{v}) - R_D^\delta(\hat{S}|\boldsymbol{v}) \\
&= R_D^\delta(S_D^\star|\boldsymbol{v}) - R_D^\delta(\hat{S}_D|\boldsymbol{v}^{\mathrm{LCB}}) + R_D^\delta(\hat{S}_D|\boldsymbol{v}^{\mathrm{LCB}}) - R_D^\delta(\hat{S}_D|\boldsymbol{v}) \\
&\leq \underbrace{R_D^\delta(S_D^\star|\boldsymbol{v}) - R_D^\delta(S_D^\star|\boldsymbol{v}^{\mathrm{LCB}})}_{(\mathrm{I})} + \underbrace{R_D^\delta(\hat{S}_D|\boldsymbol{v}^{\mathrm{LCB}}) - R_D^\delta(\hat{S}_D|\boldsymbol{v})}_{(\mathrm{II})} \qquad (17)
\end{aligned}
$$

where the last inequality holds due to the optimality of $\hat{S}_D$ with respect to $\boldsymbol{v}^{\mathrm{LCB}}$.

In the following subsections, we derive an upper bound for term (I) by leveraging its dual representation. For term (II), we establish a monotonicity property for the robust revenue to demonstrate that this term is non-positive.

### C.1. Sub-Optimality Gap under TV ambiguity Set

We begin by establishing an auxiliary lemma to analyze the sensitivity of the revenue to the utility vector $\boldsymbol{u} \in \mathbb{R}^N$, where $\boldsymbol{v} = \exp(\boldsymbol{u})$. Given a utility vector $\boldsymbol{u}$ and an assortment $S$, the choice probability for item $i \in S$ is defined as:

$$
p_i(\boldsymbol{u}) := \frac{\exp(u_i)}{1 + \sum_{j \in S} \exp(u_j)}.
$$

For any subset of items $I \subseteq S$, we define the function

$$
Q(\boldsymbol{u}) := \sum_{i \in I} \frac{\exp(u_i)}{1 + \sum_{j \in S} \exp(u_j)}.
$$

The following lemma characterizes the derivatives of $Q(\boldsymbol{u})$ and provides a Taylor-based bound on its variation. While similar results appear in Lemma E.3 and Theorem 2 of (Lee & Oh, 2024), their analysis is restricted to the case where $I = S$ and revenues are uniform ($r_i \equiv 1$). We generalize these results to arbitrary subsets $I \subseteq S$.

**Lemma C.1.** *For all $i, j \in S$, the second-order partial derivatives of $Q(\boldsymbol{u})$ satisfy*

$$
\left| \frac{\partial^2 Q}{\partial i \partial j}(\boldsymbol{u}) \right| \leq \begin{cases} p_i(\boldsymbol{u}) & \text{if } i = j \\ 2p_i(\boldsymbol{u})p_j(\boldsymbol{u}) & \text{if } i \neq j \end{cases}
$$

*Moreover, for any utility vectors $\boldsymbol{u}, \boldsymbol{u}^\star \in \mathbb{R}^N$, their difference is bounded by*

$$
Q(\boldsymbol{u}) - Q(\boldsymbol{u}^\star) \leq \sum_{i \in S} p_i(\boldsymbol{u}^\star)|u_i - u_i^\star| + \frac{3}{2} \max_{i \in S}(u_i - u_i^\star)^2.
$$

*Proof of Lemma C.1.* For any $i \in S$, the first-order derivative is given by

$$
\begin{aligned}
\frac{\partial Q}{\partial i}(\boldsymbol{u}) &= \frac{\exp(u_i)\mathbf{1}\{i \in I\}}{1 + \sum_{j \in S} \exp(u_j)} - \frac{\exp(u_i) \sum_{j \in I} \exp(u_j)}{(1 + \sum_{j \in S} \exp(u_j))^2} \\
&= p_i(\boldsymbol{u})\left(\mathbf{1}\{i \in I\} - Q(\boldsymbol{u})\right).
\end{aligned}
$$

For any $i, j \in S$, using the property $\frac{\partial p_i}{\partial u_j}(\boldsymbol{u}) = p_i(\boldsymbol{u})(\delta_{ij} - p_j(\boldsymbol{u}))$ and $\frac{\partial}{\partial u_j}(\mathbf{1}\{i \in I\} - Q(\boldsymbol{u})) = -\frac{\partial Q}{\partial u_j}(\boldsymbol{u})$, where $\delta_{ij}$ is the Kronecker delta, we have

$$
\begin{aligned}
\frac{\partial^2 Q}{\partial i \partial j}(\boldsymbol{u}) &= \frac{\partial p_i}{\partial u_j}(\boldsymbol{u}) \cdot (\mathbf{1}\{i \in I\} - Q(\boldsymbol{u})) + p_i(\boldsymbol{u}) \cdot \frac{\partial}{\partial u_j}(\mathbf{1}\{i \in I\} - Q(\boldsymbol{u})) \\
&= p_i(\boldsymbol{u})(\delta_{ij} - p_j(\boldsymbol{u}))(\mathbf{1}\{i \in I\} - Q(\boldsymbol{u})) + p_i(\boldsymbol{u})(-p_j(u)(\mathbf{1}\{j \in I\} - Q(\boldsymbol{u})) \\
&= p_i(\boldsymbol{u})(\mathbf{1}\{i \in I\} - Q(\boldsymbol{u}))\delta_{ij} - p_i(\boldsymbol{u})p_j(\boldsymbol{u})(\mathbf{1}\{i \in I\} + \mathbf{1}\{j \in I\} - 2Q(\boldsymbol{u})).
\end{aligned}
$$

If $i = j$, since $p_i(\boldsymbol{u}) \in [0, 1]$ and $|\mathbf{1}\{i \in I\} - Q(\boldsymbol{u})| \le 1$, we have

$$\frac{\partial^2 Q}{\partial i^2}(\boldsymbol{u}) = p_i(\boldsymbol{u})(\mathbf{1}\{i \in I\} - Q(\boldsymbol{u})) - 2p_i(\boldsymbol{u})p_i(\boldsymbol{u})\left(\mathbf{1}\{i \in I\} - Q(\boldsymbol{u})\right)$$

$$= p_i(\boldsymbol{u})(1 - 2p_i(\boldsymbol{u}))(\mathbf{1}\{i \in I\} - Q(\boldsymbol{u}))$$

$$\implies \left|\frac{\partial^2 Q}{\partial i^2}(\boldsymbol{u})\right| \le p_i(\boldsymbol{u}).$$

If $i \ne j$, since $p_i(\boldsymbol{u}), p_j(\boldsymbol{u}) \in [0, 1]$ and $|\mathbf{1}\{i \in I\} + \mathbf{1}\{j \in I\} - 2Q(\boldsymbol{u})| \le 2$, we have

$$\frac{\partial^2 Q}{\partial i \partial j}(\boldsymbol{u}) = -p_i(\boldsymbol{u})p_j(\boldsymbol{u})\left(\mathbf{1}\{i \in I\} + \mathbf{1}\{j \in I\} - 2Q(\boldsymbol{u})\right)$$

$$\implies \left|\frac{\partial^2 Q}{\partial i \partial j}(\boldsymbol{u})\right| \le 2p_i(\boldsymbol{u})p_j(\boldsymbol{u}).$$

Then, we apply the second-order Taylor expansion

$$Q(\boldsymbol{u}) - Q(\boldsymbol{u}^\star) = \nabla Q(\boldsymbol{u}^\star)^\top (\boldsymbol{u} - \boldsymbol{u}^\star) + \frac{1}{2}(\boldsymbol{u} - \boldsymbol{u}^\star)^\top \nabla^2 Q(\bar{\boldsymbol{u}})(\boldsymbol{u} - \boldsymbol{u}^\star), \tag{18}$$

where $\bar{\boldsymbol{u}}$ is a convex combination of $\boldsymbol{u}$ and $\boldsymbol{u}^\star$.

The first term in Equation (18) is bounded by,

$$\nabla Q(\boldsymbol{u}^\star)^\top (\boldsymbol{u} - \boldsymbol{u}^\star) = \sum_{i \in S} p_i(\boldsymbol{u}^\star) \left(\mathbf{1}\{i \in I\} - Q(\boldsymbol{u}^\star)\right)(u_i - u_i^\star)$$

$$\le \sum_{i \in S} p_i(\boldsymbol{u}^\star)|u_i - u_i^\star|$$

Using the derivative bounds derived above, the second term in Equation (18) is bounded by,

$$\frac{1}{2}(\boldsymbol{u} - \boldsymbol{u}^\star)^\top \nabla^2 Q(\bar{\boldsymbol{u}})(\boldsymbol{u} - \boldsymbol{u}^\star) = \frac{1}{2}\sum_{i \in S}\sum_{j \in S}(u_i - u_i^\star)\frac{\partial^2 Q}{\partial i \partial j}(\bar{\boldsymbol{u}})(u_j - u_j^\star)$$

$$= \frac{1}{2}\sum_{i \in S}\sum_{j \in S, j \ne i}(u_i - u_i^\star)\frac{\partial^2 Q}{\partial i \partial j}(\bar{\boldsymbol{u}})(u_j - u_j^\star) + \frac{1}{2}\sum_{i \in S}(u_i - u_i^\star)^2\frac{\partial^2 Q}{\partial i^2}(\bar{\boldsymbol{u}})$$

$$\le \sum_{i \in S}\sum_{j \in S, j \ne i} p_i(\bar{\boldsymbol{u}})p_j(\bar{\boldsymbol{u}})|u_i - u_i^\star||u_j - u_j^\star| + \frac{1}{2}\sum_{i \in S} p_i(\bar{\boldsymbol{u}})(u_i - u_i^\star)^2$$

$$\le \sum_{i \in S}\sum_{j \in S} p_i(\bar{\boldsymbol{u}})p_j(\bar{\boldsymbol{u}})|u_i - u_i^\star||u_j - u_j^\star| + \frac{1}{2}\sum_{i \in S} p_i(\bar{\boldsymbol{u}})(u_i - u_i^\star)^2$$

$$\le \frac{1}{2}\sum_{i \in S}\sum_{j \in S} p_i(\bar{\boldsymbol{u}})p_j(\bar{\boldsymbol{u}})(u_i - u_i^\star)^2 + \frac{1}{2}\sum_{i \in S}\sum_{j \in S} p_i(\bar{\boldsymbol{u}})p_j(\bar{\boldsymbol{u}})(u_j - u_j^\star)^2 + \frac{1}{2}\sum_{i \in S} p_i(\bar{\boldsymbol{u}})(u_i - u_i^\star)^2$$

$$\le \frac{3}{2}\sum_{i \in S} p_i(\bar{\boldsymbol{u}})(u_i - u_i^\star)^2$$

$$\le \frac{3}{2}\max_{i \in S}(u_i - u_i^\star)^2$$

By combing these bounds, we complete the proof

$$Q(\boldsymbol{u}) - Q(\boldsymbol{u}^\star) \le \sum_{i \in S} p_i(\boldsymbol{u}^\star)|u_i - u_i^\star| + \frac{3}{2}\max_{i \in S}(u_i - u_i^\star)^2.$$

$$\square$$

We now establish the sub-optimality gap under the TV ambiguity set.

***Proof of Theorem 4.5.*** For the KL ambiguity set, given $\boldsymbol{\theta}^\star$, we adopt the following shorthand notation for convenience: $p(\cdot|S) := p(\cdot|S, \boldsymbol{\theta}^\star)$, $\boldsymbol{v} := \exp(\boldsymbol{X}^\top \boldsymbol{\theta}^\star)$, $S^\star := S_{\text{TV}}^\star$ and $\hat{S} := \hat{S}_{\text{TV}}$.

**Bounding Term (I).** We first sort the rewards within assortment $S^\star$ as $r_1 \leq r_2 \leq \cdots \leq r_{|S^\star|}$, and define the indices

$$k := \min\left\{ j \in S^\star : \sum_{i \in S : i \geq j} p(i|S^\star) \leq \delta \right\} \quad \text{and} \quad k' = \min\left\{ j \in S^\star : \sum_{i \in S : i \geq j} \hat{p}(i|S^\star) \leq \delta \right\}.$$

Using the dual representation in Equation (14), we have

$$R_{\text{TV}}^\delta(S^\star|\boldsymbol{v}) = \Phi(r_k; S^\star, \boldsymbol{v}) \quad \text{and} \quad R_{\text{TV}}^\delta(S^\star|\boldsymbol{v}^{\text{LCB}}) = \Phi(r_{k'}; S^\star, \boldsymbol{v}^{\text{LCB}}) \geq \Phi(r_k; S^\star, \boldsymbol{v}^{\text{LCB}}).$$

Since $r_i \in [0, 1]$ and $0 \leq r_k - r_i \leq 1$ for all $i < k$, it follows that

$$\begin{aligned}
R_{\text{TV}}^\delta(S^\star|\boldsymbol{v}) - R_{\text{TV}}^\delta(S^\star|\boldsymbol{v}^{\text{LCB}}) &= \Phi(r_k; S^\star, \boldsymbol{v}) - \Phi(r_{k'}; S^\star, \boldsymbol{v}^{\text{LCB}}) \\
&\leq \Phi(r_k; S^\star, \boldsymbol{v}) - \Phi(r_k; S^\star, \boldsymbol{v}^{\text{LCB}}) \\
&= \sum_{i \in S^\star : i < k} (\hat{p}(i|S^\star) - p(i|S^\star)) \cdot (r_k - r_i) \\
&\leq \sum_{i \in I} (\hat{p}(i|S^\star) - p(i|S^\star)) \\
&\leq \sum_{i \in S^\star} p(i|S^\star)|u_i - u_i^{\text{LCB}}| + \frac{3}{2} \max_{i \in S^\star}(u_i - u_i^{\text{LCB}})^2,
\end{aligned} \quad (19)$$

where $I := \{i \in S^\star : i < k, \hat{p}(i|S^\star) > p(i|S^\star)\}$, and the last inequality follows from Lemma C.1.

**Bounding Term (II).** We next demonstrate that Term (II) is non-positive. From Lemma 3.1, the difference in robust rewards is expressed as

$$\begin{aligned}
&R_{\text{TV}}^\delta(\hat{S}|\boldsymbol{v}^{\text{LCB}}) - R_{\text{TV}}^\delta(\hat{S}|\boldsymbol{v}) \\
&= \max_{\alpha \in [0, 1/\delta]} \left\{ -\mathbb{E}_{i \sim \hat{p}(\cdot|\hat{S})}\left[ (\alpha - r_i)_+ \right] + (1 - \delta)\alpha \right\} - \max_{\alpha \in [0, 1/\delta]} \left\{ -\mathbb{E}_{i \sim p(\cdot|\hat{S})}\left[ (\alpha - r_i)_+ \right] + (1 - \delta)\alpha \right\} \\
&\leq -\mathbb{E}_{i \sim \hat{p}(\cdot|\hat{S})}\left[ (\hat{\alpha}^\star - r_i)_+ \right] + \mathbb{E}_{i \sim p(\cdot|\hat{S})}\left[ (\hat{\alpha}^\star - r_i)_+ \right] \\
&= \sum_{i \in \hat{S}_+} p(i|\hat{S})(\hat{\alpha}^\star - r_i)_+ - \sum_{i \in \hat{S}_+} \hat{p}(i|\hat{S})(\hat{\alpha}^\star - r_i)_+ \\
&= \frac{\sum_{i \in \hat{S}_+} v_i (\hat{\alpha}^\star - r_i)_+}{\sum_{i \in \hat{S}_+} v_i} - \frac{\sum_{i \in \hat{S}_+} v_i^{\text{LCB}} (\hat{\alpha}^\star - r_i)_+}{\sum_{i \in \hat{S}_+} v_i^{\text{LCB}}},
\end{aligned}$$

where $\hat{\alpha}^\star = \operatorname{argmax}_{\alpha \in [0, 1/\delta]} \left\{ -\mathbb{E}_{i \sim \hat{p}(\cdot|\hat{S})}\left[ (\alpha - r_i)_+ \right] + (1 - \delta)\alpha \right\}$ is the optimizer for the LCB-based robust reward.

To show that this difference is non-positive, it suffices to prove

$$\frac{\sum_{i \in \hat{S}_+} v_i (\hat{\alpha}^\star - r_i)_+}{\sum_{i \in \hat{S}_+} v_i} \leq \frac{\sum_{i \in \hat{S}_+} v_i^{\text{LCB}} (\hat{\alpha}^\star - r_i)_+}{\sum_{i \in \hat{S}_+} v_i^{\text{LCB}}}.$$

By Lemma C.2, any item $j$ belongs to assortment $\hat{S}$ must satisfy

$$r_j \geq R_{\text{TV}}^\delta(\hat{S}|\boldsymbol{v}^{\text{LCB}}) + \delta\hat{\alpha}^\star = -\mathbb{E}_{i \sim \hat{p}(\cdot|\hat{S})}\left[ (\hat{\alpha}^\star - r_i)_+ \right] + \hat{\alpha}^\star,$$

which is equivalent to

$$\hat{\alpha}^\star - r_j \leq \frac{\sum_{i \in \hat{S}_+} v_i^{\text{LCB}} (\hat{\alpha}^\star - r_i)_+}{\sum_{i \in \hat{S}_+} v_i^{\text{LCB}}} \implies (\hat{\alpha}^\star - r_j)_+ \leq \frac{\sum_{i \in \hat{S}_+} v_i^{\text{LCB}} (\hat{\alpha}^\star - r_i)_+}{\sum_{i \in \hat{S}_+} v_i^{\text{LCB}}}.$$

Using the inequality above, we have

$$
\begin{aligned}
\frac{\sum_{i \in \hat{S}_+} v_i \left(\hat{\alpha}^\star - r_i\right)_+}{\sum_{i \in \hat{S}_+} v_i} &= \frac{\sum_{i \in \hat{S}_+} v_i^{\mathrm{LCB}} \left(\hat{\alpha}^\star - r_i\right)_+ + \sum_{i \in \hat{S}_+} (v_i - v_i^{\mathrm{LCB}}) \left(\hat{\alpha}^\star - r_i\right)_+}{\sum_{i \in \hat{S}_+} v_i^{\mathrm{LCB}} + \sum_{i \in \hat{S}_+} v_i - v_i^{\mathrm{LCB}}} \\
&\leq \frac{\sum_{i \in \hat{S}_+} v_i^{\mathrm{LCB}} \left(\hat{\alpha}^\star - r_i\right)_+ + \sum_{i \in \hat{S}_+} (v_i - v_i^{\mathrm{LCB}}) \frac{\sum_{i \in \hat{S}} v_i^{\mathrm{LCB}} (\hat{\alpha}^\star - r_i)_+}{\sum_{i \in \hat{S}} v_i^{\mathrm{LCB}}}}{\sum_{i \in \hat{S}_+} v_i^{\mathrm{LCB}} + \sum_{i \in \hat{S}_+} v_i - v_i^{\mathrm{LCB}}} \\
&= \frac{\sum_{i \in \hat{S}_+} v_i^{\mathrm{LCB}} \left(\hat{\alpha}^\star - r_i\right)_+}{\sum_{i \in \hat{S}_+} v_i^{\mathrm{LCB}}},
\end{aligned}
$$

where the last equality follows from the equation that $(a + c \cdot (a/b))/(b + c) = a/b$.

Consequently, we establish that

$$
R_{\mathrm{TV}}^\delta(\hat{S}|\boldsymbol{v}^{\mathrm{LCB}}) - R_{\mathrm{TV}}^\delta(\hat{S}|\boldsymbol{v}) \leq 0. \tag{20}
$$

**Final Bound.** Combining the upper bound for term (I) from Equation (19) and the non-positivity of term (II) from Equation (20), and substituting the utility gap from Equation (16), we obtain that

$$
\begin{aligned}
\mathrm{SubOpt}_{\mathrm{TV}}^\delta(\hat{S}|\boldsymbol{\theta}^\star) &\leq \sum_{i \in S^\star} p(i|S^\star)|u_i - u_i^{\mathrm{LCB}}| + \frac{3}{2} \max_{i \in S^\star}(u_i - u_i^{\mathrm{LCB}})^2 \\
&\leq 32 \sum_{i \in S^\star} p(i|S^\star)\|\boldsymbol{x}_i\|_{\boldsymbol{H}_T^\lambda(\boldsymbol{\theta}^\star)^{-1}} \left(\sqrt{\log(N/\zeta)} + \sqrt{\lambda}W\right) + 1536 \max_{i \in S^\star} \|\boldsymbol{x}_i\|_{\boldsymbol{H}_T^\lambda(\boldsymbol{\theta}^\star)^{-1}}^2 \left(\sqrt{\log(N/\zeta)} + \sqrt{\lambda}W\right)^2
\end{aligned}
$$

$\square$

**Lemma C.2.** *Consider the robust revenue under TV ambiguity set with a fixed attraction vector $\boldsymbol{v}$. If an item $j \in [N]$ belongs to the optimal robust assortment $S_{\mathrm{TV}}^\star$, then its reward $r_j$ must satisfy:*

$$
r_j \geq R_{\mathrm{TV}}^\delta(S_{\mathrm{TV}}^\star|\boldsymbol{v}) + \delta\alpha^\star, \quad (S_{\mathrm{TV}}^\star, \alpha^\star) \in \operatorname*{argmax}_{S \in \mathcal{S}_K, \alpha \in [0, 1/\delta]} \Phi(\alpha; S, \boldsymbol{v}) \text{ with } j \in S_{\mathrm{TV}}^\star.
$$

*where $\Phi(\alpha; S, \boldsymbol{v}) := -\mathbb{E}_{i \sim p(\cdot|S, \boldsymbol{v})}\left[(\alpha - r_i)_+\right] + (1 - \delta)\alpha$.*

*Proof of Lemma C.2.* We prove it by contradiction. If there exists an item $j$ such that $j \in S_{\mathrm{TV}}^\star$ for some $(S_{\mathrm{TV}}^\star, \alpha^\star)$ satisfying $(S_{\mathrm{TV}}^\star, \alpha^\star) \in \operatorname{argmax}_{S \in \mathcal{S}_K, \alpha \in [0, 1/\delta]} \Phi(\alpha; S, \boldsymbol{v})$, but

$$
r_j < R_{\mathrm{TV}}^\delta(S_{\mathrm{TV}}^\star|\boldsymbol{v}) + \delta\alpha^\star \quad \Leftrightarrow \quad (\alpha^\star - r_j)_+ > \frac{\sum_{i \in S_{\mathrm{TV}}^\star} v_i(\alpha^\star - r_i)_+}{\sum_{i \in S_{\mathrm{TV}}^\star} v_i} \tag{21}
$$

Then considering the new assortment $S_{\mathrm{TV}}^\star \setminus \{j\} \in \mathcal{S}_K$. Let $A := \sum_{i \in S_{\mathrm{TV}}^\star} v_i(\alpha^\star - r_i)_+$ and $B := \sum_{i \in S_{\mathrm{TV}}^\star} v_i$. Note that

$$
\Phi(\alpha^\star; S_{\mathrm{TV}}^\star \setminus \{j\}, \boldsymbol{v}) > \Phi(\alpha^\star; S_{\mathrm{TV}}^\star, \boldsymbol{v}) \quad \Leftrightarrow \quad \frac{A - v_j(\alpha^\star - r_j)_+}{B - v_j} < \frac{A}{B} \quad \Leftrightarrow \quad (\alpha^\star - r_j)_+ > \frac{A}{B}.
$$

The condition in (21) implies that $(\alpha^\star - r_j)_+ > A/B$, which means that $\Phi(\alpha^\star; S_{\mathrm{TV}}^\star \setminus \{j\}, \boldsymbol{v})$. Therefore, $S_{\mathrm{TV}}^\star \setminus \{j\}$ yields a higher objective value than $S_{\mathrm{TV}}^\star$ for the same $\alpha^\star$, which contradicts the optimality of $S_{\mathrm{TV}}^\star$.

By calculation, we can obtain that $\Phi(\alpha^\star; S_{\mathrm{TV}}^\star \setminus \{j\}, \boldsymbol{v}) > \Phi(\alpha^\star; S_{\mathrm{TV}}^\star, \boldsymbol{v})$, which contradicts (21). Thus we must have $r_j \geq R_{\mathrm{TV}}^\delta(S_{\mathrm{TV}}^\star|\boldsymbol{v}) + \delta\alpha^\star$. $\square$

## C.2. Sub-Optimality Gap under KL ambiguity Set

***Proof of Theorem 4.6.*** For the KL ambiguity set, given $\boldsymbol{\theta}^\star$, we adopt the following shorthand notation for convenience: $p(\cdot|S) := p(\cdot|S, \boldsymbol{\theta}^\star)$, $\boldsymbol{v} := \exp(\boldsymbol{X}^\top \boldsymbol{\theta}^\star)$, $S^\star := S^\star_{\mathrm{KL}}$ and $\hat{S} := \hat{S}_{\mathrm{KL}}$.

**Bounding Term (I).** Utilizing the dual formulation established in Lemma 3.3, the robust revenue for a fixed assortment $S$ is given by

$$R^\delta_{\mathrm{KL}}(S|\boldsymbol{v}) = \max_{0 \le \rho \le 1/\delta} \left\{ -\rho \log \left( \mathbb{E}_{i \sim p(\cdot|S,\boldsymbol{v})}[\exp(-r_i/\rho)] \right) - \rho\delta \right\}$$

$$= \max_{0 \le \rho \le 1/\delta} \left\{ -\rho \log \left( \frac{\sum_{i \in S_+} v_i \cdot \exp(-r_i/\rho)}{\sum_{i \in S_+} v_i} \right) - \rho\delta \right\}.$$

Term (I) under KL ambiguity set can be bounded by

$$R^\delta_{\mathrm{KL}}(S^\star|\boldsymbol{v}) - R^\delta_{\mathrm{KL}}(S^\star|\boldsymbol{v}^{\mathrm{LCB}}) \le \max_{0 \le \rho \le 1/\delta} \rho \cdot \left[ \log \left( \frac{\sum_{i \in S^\star_+} v_i^{\mathrm{LCB}} \exp(-r_i/\rho)}{\sum_{j \in S^\star_+} v_j^{\mathrm{LCB}}} \right) - \log \left( \frac{\sum_{i \in S^\star_+} v_i \exp(-r_i/\rho)}{\sum_{j \in S^\star_+} v_j} \right) \right]$$

$$\le \max_{0 \le \rho \le 1/\delta} \rho \cdot \left( \frac{\sum_{i \in S^\star_+} v_i^{\mathrm{LCB}} \exp(-r_i/\rho)}{\sum_{j \in S^\star_+} v_j^{\mathrm{LCB}}} \cdot \frac{\sum_{j \in S^\star_+} v_j}{\sum_{i \in S^\star_+} v_i \exp(-r_i/\rho)} - 1 \right)$$

$$\le \min \left\{ \frac{1}{\log(3/2)}, \frac{1}{\delta} \right\} \sum_{i \in S^\star_+} \frac{v_i - v_i^{\mathrm{LCB}}}{\sum_{j \in S^\star_+} v_j^{\mathrm{LCB}}},$$

where the first inequality holds due to the property $\max f - \max g \le \max(f - g)$, the second inequality follows from the inequality that $\log x \le x - 1$ for all $x > 0$, and the last inequality follows Lemma C.3.

Furthermore, by applying the utility gap bound from (16), the sum above can be bounded by

$$\sum_{i \in S^\star_+} \frac{v_i - v_i^{\mathrm{LCB}}}{\sum_{j \in S^\star_+} v_j^{\mathrm{LCB}}} = \sum_{i \in S^\star_+} \frac{v_i^{\mathrm{LCB}} \exp(u_i - u_i^{\mathrm{LCB}}) - v_i^{\mathrm{LCB}}}{\sum_{j \in S^\star_+} v_j^{\mathrm{LCB}}}$$

$$= \sum_{i \in S^\star_+} \hat{p}(i|S^\star)(\exp(u_i - u_i^{\mathrm{LCB}}) - 1)$$

$$\le \sum_{i \in S^\star_+} \hat{p}(i|S^\star) \left[ \exp \left( 32\|\boldsymbol{x}_i\|_{\boldsymbol{H}_T^\lambda(\boldsymbol{\theta}^\star)^{-1}} \left( \sqrt{\log(N/\zeta)} + \sqrt{\lambda}W \right) \right) - 1 \right]$$

$$\le 32 \sum_{i \in S^\star_+} \hat{p}(i|S^\star_+)\|\boldsymbol{x}_i\|_{\boldsymbol{H}_T^\lambda(\boldsymbol{\theta}^\star)^{-1}} \left( \sqrt{\log(N/\zeta)} + \sqrt{\lambda}W \right).$$

where the last inequality holds because $e^x - 1 \le x$ when $x \in [0, 1]$, assuming the confidence radius satisfies the small-gap condition $32\|\boldsymbol{x}_i\|_{\boldsymbol{H}_T^\lambda(\boldsymbol{\theta}^\star)^{-1}} \left( \sqrt{\log(N/\zeta)} + \sqrt{\lambda}W \right) \le 1$.

In addition, based on this small-gap condition, we have

$$\hat{p}(i|S^\star_+) = \frac{v_i^{\mathrm{LCB}}}{\sum_{j \in S^\star_+} v_j^{\mathrm{LCB}}} \le \frac{v_i}{\sum_{j \in S^\star_+} v_j \exp(u_j^{\mathrm{LCB}} - u_j)} \le e \cdot \frac{v_i}{\sum_{j \in S^\star_+} v_j} = e \cdot p(i|S^\star_+).$$

**Bounding Term (II).** *We can show that term (II) is non-positive by Lemma D.1 in Lu et al. (2026), and we repeat their arguments here just for completeness.* By Lemma 3.3, we can bound term (II) as

$$R^\delta_{\mathrm{KL}}(\hat{S}|\boldsymbol{v}^{\mathrm{LCB}}) - R^\delta_{\mathrm{KL}}(\hat{S}|\boldsymbol{v})$$

$$= \max_{0 \le \rho \le 1/\delta} \left\{ -\rho \log \left( \mathbb{E}_{i \sim \hat{p}(\cdot|\hat{S})}[\exp(-r_i/\rho)] \right) - \rho\delta \right\} - \max_{0 \le \rho \le 1/\delta} \left\{ -\rho \log \left( \mathbb{E}_{i \sim p(\cdot|\hat{S})}[\exp(-r_i/\rho)] \right) - \rho\delta \right\}$$

$$\le \hat{\rho}^\star \log \left( \mathbb{E}_{i \sim p(\cdot|\hat{S})}[\exp(-r_i/\hat{\rho}^\star)] \right) - \hat{\rho}^\star \log \left( \mathbb{E}_{i \sim \hat{p}(\cdot|\hat{S})}[\exp(-r_i/\hat{\rho}^\star)] \right),$$

where $\hat{\rho}^\star := \text{argmax}_{0 \leq \rho \leq 1/\delta} \left\{ -\rho \log \left( \mathbb{E}_{i \sim \hat{p}(\cdot | \hat{S})}[\exp(-r_i/\rho)] \right) - \rho\delta \right\}$.

To show that this difference is non-positive, it suffices to prove

$$\frac{\sum_{i \in \hat{S}_+} v_i \exp(-r_i/\hat{\rho}^\star)}{\sum_{i \in \hat{S}_+} v_i} \leq \frac{\sum_{i \in \hat{S}_+} v_i^{\text{LCB}} \exp(-r_i/\hat{\rho}^\star)}{\sum_{i \in \hat{S}_+} v_i^{\text{LCB}}}.$$

By Lemma C.4, any item $j$ belongs to assortment $\hat{S}$ must satisfy

$$r_j \geq R_{\text{KL}}^\delta(\hat{S}|\boldsymbol{v}^{\text{LCB}}) + \delta\hat{\rho}^\star = -\hat{\rho}^\star \log \left( \mathbb{E}_{i \sim \hat{p}(\cdot | \hat{S})}[\exp(-r_i/\hat{\rho}^\star)] \right),$$

which is equivalent to

$$\exp(-r_j/\hat{\rho}^\star) \leq \frac{\sum_{i \in \hat{S}_+} v_i^{\text{LCB}} \exp(-r_i/\hat{\rho}^\star)}{\sum_{i \in \hat{S}_+} v_i^{\text{LCB}}}.$$

Using the inequality above, we have

$$\frac{\sum_{i \in \hat{S}_+} v_i \exp(-r_i/\hat{\rho}^\star)}{\sum_{i \in \hat{S}_+} v_i} = \frac{\sum_{i \in \hat{S}_+} v_i^{\text{LCB}} \exp(-r_i/\hat{\rho}^\star) + \sum_{i \in \hat{S}_+} (v_i - v_i^{\text{LCB}}) \exp(-r_i/\hat{\rho}^\star)}{\sum_{i \in \hat{S}_+} v_i^{\text{LCB}} + \sum_{i \in \hat{S}_+} v_i - v_i^{\text{LCB}}}$$

$$\leq \frac{\sum_{i \in \hat{S}_+} v_i^{\text{LCB}} \exp(-r_i/\hat{\rho}^\star) + \sum_{i \in \hat{S}_+} (v_i - v_i^{\text{LCB}}) \frac{\sum_{i \in \hat{S}} v_i^{\text{LCB}} \exp(-r_i/\hat{\rho}^\star)}{\sum_{i \in \hat{S}} v_i^{\text{LCB}}}}{\sum_{i \in \hat{S}_+} v_i^{\text{LCB}} + \sum_{i \in \hat{S}_+} v_i - v_i^{\text{LCB}}}$$

$$= \frac{\sum_{i \in \hat{S}_+} v_i^{\text{LCB}} \exp(-r_i/\hat{\rho}^\star)}{\sum_{i \in \hat{S}_+} v_i^{\text{LCB}}},$$

where the last equality follows from the equation that $(a + c \cdot (a/b))/(b + c) = a/b$.

Consequently, we establish that

$$R_{\text{KL}}^\delta(\hat{S}|\boldsymbol{v}^{\text{LCB}}) - R_{\text{KL}}^\delta(\hat{S}|\boldsymbol{v}) \leq 0.$$

**Final Bound.** Combining the results for term (I) and term (II) yields:

$$\text{SubOpt}_{\text{KL}}^\delta(\hat{S}|\boldsymbol{\theta}^\star) \leq 32e \cdot \min\left\{ \frac{1}{\log(3/2)}, \frac{1}{\delta} \right\} \sum_{i \in S^\star} p(i|S^\star) \|\boldsymbol{x}_i\|_{\boldsymbol{H}_T^\lambda(\boldsymbol{\theta}^\star)^{-1}} \left( \sqrt{\log(N/\zeta)} + \sqrt{\lambda}W \right).$$

$\square$

*We introduce the following Lemma from Lu et al. (2026), where the proof is presented just for completeness.*

**Lemma C.3** (Theorem B.1 in Lu et al. (2026)).

$$\max_{0 \leq \rho \leq 1/\delta} \rho \cdot \left( \frac{\sum_{i \in S_+^\star} v_i^{\text{LCB}} \exp(-r_i/\rho)}{\sum_{j \in S_+^\star} v_j^{\text{LCB}}} \cdot \frac{\sum_{j \in S_+^\star} v_j}{\sum_{i \in S_+^\star} v_i \exp(-r_i/\rho)} - 1 \right) \leq \min\left\{ \frac{1}{\log(3/2)}, \frac{1}{\delta} \right\} \sum_{i \in S_+^\star} \frac{v_i - v_i^{\text{LCB}}}{\sum_{j \in S_+^\star} v_j^{\text{LCB}}}.$$

***Proof of Lemma C.3.*** To derive a tight upper bound on LHS, we consider two ranges of the dual variable $\rho$ separately. More specifically, we consider

$$\text{LHS} \leq \max \left\{ \underbrace{\max_{0 \leq \rho \leq c} \left\{ \rho \cdot \left( \frac{\sum_{i \in S_+^\star} v_i^{\text{LCB}} \cdot \exp(-r_i/\rho)}{\sum_{i \in S_+^\star} v_i^{\text{LCB}}} \cdot \frac{\sum_{i \in S_+^\star} v_i}{\sum_{i \in S_+^\star} v_i \cdot \exp(-r_i/\rho)} - 1 \right) \right\}}_{\text{Term (i)}}, \right.$$

$$\underbrace{\max_{c \leq \rho \leq 1/\delta} \left\{ \rho \cdot \left( \frac{\sum_{i \in S_+^\star} v_i^{\mathrm{LCB}} \cdot \exp(-r_i/\rho)}{\sum_{i \in S_+^\star} v_i^{\mathrm{LCB}}} \cdot \frac{\sum_{i \in S_+^\star} v_i}{\sum_{i \in S_+^\star} v_i \cdot \exp(-r_i/\rho)} - 1 \right) \right\}}_{\text{Term (ii)}},$$

where $c > 0$ is a parameter to be tuned later. We now upper bound Term (i) and Term (ii) separately.

**Upper bounding Term (i).** To upper bound Term (i), we utilize the inequality that

$$\sum_i a_i = \sum_i \frac{a_i}{b_i} \cdot b_i \leq \max_i \left\{ \frac{a_i}{b_i} \right\} \cdot \sum_i b_i, \quad \forall a_i \in \mathbb{R}, \ b_i > 0, \tag{22}$$

which gives that for any fixed dual variable $0 \leq \rho \leq c$, we have the following inequalities,

$$\frac{\sum_{i \in S_+^\star} v_i^{\mathrm{LCB}} \cdot \exp(-r_i/\rho)}{\sum_{i \in S_+^\star} v_i^{\mathrm{LCB}}} \cdot \frac{\sum_{i \in S_+^\star} v_i}{\sum_{i \in S_+^\star} v_i \cdot \exp(-r_i/\rho)} - 1$$

$$= \frac{\sum_{i \in S_+^\star} v_i}{\sum_{i \in S_+^\star} v_i \cdot \exp(-r_i/\rho)} \cdot \left( \sum_{i \in S_+^\star} \left( \frac{v_i^{\mathrm{LCB}}}{\sum_{i \in S_+^\star} v_i^{\mathrm{LCB}}} - \frac{v_i}{\sum_{i \in S_+^\star} v_i} \right) \cdot \exp(-r_i/\rho) \right)$$

$$\leq \frac{\sum_{i \in S_+^\star} v_i}{\sum_{i \in S_+^\star} v_i \cdot \exp(-r_i/\rho)} \cdot \max_{i \in S_+^\star} \left\{ \left( \frac{v_i^{\mathrm{LCB}}}{\sum_{i \in S_+^\star} v_i^{\mathrm{LCB}}} - \frac{v_i}{\sum_{i \in S_+^\star} v_i} \right) \cdot \frac{\sum_{i \in S_+^\star} v_i}{v_i} \right\} \cdot \frac{\sum_{i \in S_+^\star} v_i \cdot \exp(-r_i/\rho)}{\sum_{i \in S_+^\star} v_i}$$

$$= \max_{i \in S_+^\star} \left\{ \left( \frac{v_i^{\mathrm{LCB}}}{\sum_{i \in S_+^\star} v_i^{\mathrm{LCB}}} - \frac{v_i}{\sum_{i \in S_+^\star} v_i} \right) \cdot \frac{\sum_{i \in S_+^\star} v_i}{v_i} \right\}, \tag{23}$$

where the inequality uses (22). To further upper bound the right hand side of (23), we utilize the pessimistic property of $\boldsymbol{v}^{\mathrm{LCB}}$ to derive that for any item $i \in S_+^\star$,

$$\left( \frac{v_i^{\mathrm{LCB}}}{\sum_{i \in S_+^\star} v_i^{\mathrm{LCB}}} - \frac{v_i}{\sum_{i \in S_+^\star} v_i} \right) \cdot \frac{\sum_{i \in S_+^\star} v_i}{v_i} \leq \left( \frac{1}{\sum_{i \in S_+^\star} v_i^{\mathrm{LCB}}} - \frac{1}{\sum_{i \in S_+^\star} v_i} \right) \cdot \sum_{i \in S_+^\star} v_i = \frac{\sum_{i \in S_+^\star} v_i - v_i^{\mathrm{LCB}}}{\sum_{i \in S_+^\star} v_i^{\mathrm{LCB}}}.$$

Therefore, we can upper bound Term (i) via the following,

$$\text{Term (i)} \leq c \cdot \frac{\sum_{i \in S_+^\star} v_i - v_i^{\mathrm{LCB}}}{\sum_{i \in S_+^\star} v_i^{\mathrm{LCB}}}. \tag{24}$$

**Upper bounding Term (ii).** We consider the following approach,

$$\frac{\sum_{i \in S_+^\star} v_i^{\mathrm{LCB}} \cdot \exp(-r_i/\rho)}{\sum_{i \in S_+^\star} v_i^{\mathrm{LCB}}} \cdot \frac{\sum_{i \in S_+^\star} v_i}{\sum_{i \in S_+^\star} v_i \cdot \exp(-r_i/\rho)} - 1$$

$$\leq \frac{\sum_{i \in S_+^\star} v_i}{\sum_{i \in S_+^\star} v_i \cdot \exp(-r_i/\rho)} \cdot \left| \sum_{i \in S_+^\star} \left( \frac{v_i^{\mathrm{LCB}}}{\sum_{i \in S_+^\star} v_i^{\mathrm{LCB}}} - \frac{v_i}{\sum_{i \in S_+^\star} v_i} \right) \cdot \exp(-r_i/\rho) \right|$$

$$= \frac{\sum_{i \in S_+^\star} v_i}{\sum_{i \in S_+^\star} v_i \cdot \exp(-r_i/\rho)} \cdot \left| \sum_{i \in S_+^\star} \left( \frac{v_i^{\mathrm{LCB}}}{\sum_{i \in S_+^\star} v_i^{\mathrm{LCB}}} - \frac{v_i}{\sum_{i \in S_+^\star} v_i} \right) \cdot \left( 1 - \exp(-r_i/\rho) \right) \right|$$

$$\leq \exp(1/\rho) \cdot \left( 1 - \exp(-1/\rho) \right) \cdot \sum_{i \in S_+^\star} \left| \frac{v_i^{\mathrm{LCB}}}{\sum_{i \in S_+^\star} v_i^{\mathrm{LCB}}} - \frac{v_i}{\sum_{i \in S_+^\star} v_i} \right|, \tag{25}$$

where the first inequality just uses $x \leq |x|$, while the second inequality uses $|\sum_i a_i b_i| \leq \max_i |b_i| \cdot \sum_i |a_i|$. We further upper bound the right handside of (25) as following. On the one hand, using the property that $\boldsymbol{v}^{\mathrm{LCB}} \leq \boldsymbol{v}$, we can derive that

$$\frac{v_i^{\mathrm{LCB}}}{\sum_{i \in S_+^\star} v_i^{\mathrm{LCB}}} - \frac{v_i}{\sum_{i \in S_+^\star} v_i} \leq \frac{v_i}{\sum_{i \in S_+^\star} v_i^{\mathrm{LCB}}} - \frac{v_i}{\sum_{i \in S_+^\star} v_i} = \frac{v_i \cdot \left( \sum_{i \in S_+^\star} v_i - v_i^{\mathrm{LCB}} \right)}{\left( \sum_{i \in S_+^\star} v_i \right) \cdot \left( \sum_{i \in S_+^\star} v_i^{\mathrm{LCB}} \right)},$$

and that

$$\frac{v_i}{\sum_{i \in S_+^\star} v_i} - \frac{v_i^{\mathrm{LCB}}}{\sum_{i \in S_+^\star} v_i^{\mathrm{LCB}}} \leq \frac{v_i}{\sum_{i \in S_+^\star} v_i^{\mathrm{LCB}}} - \frac{v_i^{\mathrm{LCB}}}{\sum_{i \in S_+^\star} v_i^{\mathrm{LCB}}} = \frac{v_i - v_i^{\mathrm{LCB}}}{\sum_{i \in S_+^\star} v_i^{\mathrm{LCB}}},$$

which together gives that

$$\sum_{i \in S_+^\star} \left| \frac{v_i^{\mathrm{LCB}}}{\sum_{i \in S_+^\star} v_i^{\mathrm{LCB}}} - \frac{v_i}{\sum_{i \in S_+^\star} v_i} \right| \leq \sum_{i \in S_+^\star} \max \left\{ \frac{v_i \cdot \left( \sum_{i \in S_+^\star} v_i - v_i^{\mathrm{LCB}} \right)}{\left( \sum_{i \in S_+^\star} v_i \right) \cdot \left( \sum_{i \in S_+^\star} v_i^{\mathrm{LCB}} \right)}, \frac{v_i - v_i^{\mathrm{LCB}}}{\sum_{i \in S_+^\star} v_i^{\mathrm{LCB}}} \right\}$$

$$\leq \sum_{i \in S_+^\star} \frac{v_i \cdot \left( \sum_{i \in S_+^\star} v_i - v_i^{\mathrm{LCB}} \right)}{\left( \sum_{i \in S_+^\star} v_i \right) \cdot \left( \sum_{i \in S_+^\star} v_i^{\mathrm{LCB}} \right)} + \frac{v_i - v_i^{\mathrm{LCB}}}{\sum_{i \in S_+^\star} v_i^{\mathrm{LCB}}}$$

$$\leq 2 \cdot \frac{\sum_{i \in S_+^\star} v_i - v_i^{\mathrm{LCB}}}{\sum_{i \in S_+^\star} v_i^{\mathrm{LCB}}}. \tag{26}$$

Here the last inequality uses the fact that $\max\{a, b\} \leq a + b$ for any $a \geq 0$ and $b \geq 0$. On the other hand, by calculation, it holds that

$$\rho \cdot \exp(1/\rho) \cdot \left( 1 - \exp(-1/\rho) \right) \leq c \cdot \left( \exp(1/c) - 1 \right), \tag{27}$$

Consequently, combining (26) and (27), we can bound Term (ii) by

$$\text{Term (ii)} \leq 2c \cdot \left( \exp(1/c) - 1 \right) \cdot \frac{\sum_{i \in S_+^\star} v_i - v_i^{\mathrm{LCB}}}{\sum_{i \in S_+^\star} v_i^{\mathrm{LCB}}}. \tag{28}$$

**Combining the bounds.** Now combining (24) and (28), we can derive that for any $c \in [0, 1/\delta]$,

$$\text{LHS} \leq \max \left\{ \text{Term (i)}, \text{Term (ii)} \right\} = c \cdot \max \left\{ 2, \exp(1/c) - 1 \right\} \cdot \frac{\sum_{i \in S_+^\star} v_i - v_i^{\mathrm{LCB}}}{\sum_{i \in S_+^\star} v_i^{\mathrm{LCB}}}. \tag{29}$$

In the meanwhile, by using the same argument as in bounding Term (i), i.e., (24), except that the dual variable $\rho$ is in the full range $[0, 1/\delta]$, we can obtain that

$$\text{LHS} \leq \frac{1}{\delta} \cdot \frac{\sum_{i \in S_+^\star} v_i - v_i^{\mathrm{LCB}}}{\sum_{i \in S_+^\star} v_i^{\mathrm{LCB}}}. \tag{30}$$

Consequently, by combining (29) and (30), we can obtain that

$$\text{LHS} \leq \underbrace{\min \left\{ \min_{c \in [0, 1/\delta]} \left\{ c \cdot \max \left\{ 2, \exp(1/c) - 1 \right\} \right\}, \frac{1}{\delta} \right\}}_{C(\delta)} \cdot \frac{\sum_{i \in S_+^\star} v_i - v_i^{\mathrm{LCB}}}{\sum_{i \in S_+^\star} v_i^{\mathrm{LCB}}}.$$

**Bound of $C(\delta)$.** We consider the following two case:

- If $\delta \le \log(3/2)$, i.e., $1/\delta \ge 1/\log(3/2)$, then we have

$$\max\{1, 2(\exp(1/c) - 1)\} = \begin{cases} 2(\exp(1/c) - 1), & \text{if } 0 \le c \le 1/\log(3/2), \\ 1, & \text{if } 1/\log(3/2) \le c \le 1/\delta. \end{cases}$$

It implies that

$$\min_{0 \le c \le 1/\delta} \{c \cdot \max\{1, 2(\exp(1/c) - 1)\}\} = \min\left\{ \min_{0 \le c \le 1/\log(3/2)} 2c(\exp(1/c) - 1), \min_{1/\log(3/2) \le c \le 1/\delta} c \right\} = \frac{1}{\log(3/2)}.$$

Thus, we obtain

$$C(\delta) = \min\left\{ \frac{1}{\log(3/2)}, \frac{1}{\delta} \right\} = \frac{1}{\log(3/2)}.$$

- If $\delta \ge \log(3/2)$, i.e., $1/\delta \le 1/\log(3/2)$, then we have $\max\{1, 2(\exp(1/c) - 1)\} = 2(\exp(1/c) - 1)$. It implies that

$$\min_{0 \le c \le 1/\delta} \{c \cdot \max\{1, 2(\exp(1/c) - 1)\}\} = \min_{0 \le c \le 1/\delta} 2c(\exp(1/c) - 1) = \frac{2}{\delta}(e^\delta - 1).$$

Thus, we have

$$C(\delta) = \min\left\{ \frac{2}{\delta}(e^\delta - 1), \frac{1}{\delta} \right\} = \frac{1}{\delta} = \min\left\{ \frac{1}{\log(3/2)}, \frac{1}{\delta} \right\}.$$

Therefore, we complete the proof. $\qquad \square$

*In the following, we introduce a lemma from Lu et al. (2026), and proof is presented just for completeness.*

**Lemma C.4** (Lemma D.2 in Lu et al. (2026)). *Consider the robust revenue under KL ambiguity set with a fixed attraction vector $\boldsymbol{v}$. If an item $j \in [N]$ belongs to the optimal robust assortment $S_{\mathrm{KL}}^\star$, then its reward $r_j$ must satisfy:*

$$r_j \ge R_{\mathrm{KL}}^\delta(S_{\mathrm{KL}}^\star | \boldsymbol{v}) + \delta \rho^\star, \quad (S_{\mathrm{KL}}^\star, \rho^\star) \in \underset{S \in \mathcal{S}_K, \rho \in [0, 1/\delta]}{\mathrm{argmax}} G(\rho; S, \boldsymbol{v}) \text{ with } j \in S_{\mathrm{KL}}^\star.$$

*where $G(\rho; S, \boldsymbol{v}) := -\rho \log \left( \mathbb{E}_{i \sim p(\cdot | S)}[\exp(-r_i/\rho)] \right) - \rho \delta.$*

*Proof of Lemma C.4.* We prove it by contradiction. If there exists an item $j$ such that $j \in S_{\mathrm{KL}}^\star$ for some $(S_{\mathrm{KL}}^\star, \rho^\star)$ satisfying $(S_{\mathrm{KL}}^\star, \rho^\star) \in \mathrm{argmax}_{S \in \mathcal{S}_K, \rho \in [0, 1/\delta]} G(\rho; S, \boldsymbol{v})$, but

$$r_j < R_{\mathrm{KL}}^\delta(S_{\mathrm{KL}}^\star | \boldsymbol{v}) + \delta \rho^\star \quad \Leftrightarrow \quad \exp(-r_j/\rho^\star) > \frac{\sum_{i \in S_{\mathrm{KL}}^\star} v_i \exp(-r_i/\rho^\star)}{\sum_{i \in S_{\mathrm{KL}}^\star} v_i} \tag{31}$$

Then considering the new assortment $S_{\mathrm{KL}}^\star \setminus \{j\} \in \mathcal{S}_K$. Let $A := \sum_{i \in S_{\mathrm{KL}}^\star} v_i \exp(-r_i/\rho^\star)$ and $B := \sum_{i \in S_{\mathrm{KL}}^\star} v_i$. Note that

$$G(\rho^\star; S_{\mathrm{KL}}^\star \setminus \{j\}, \boldsymbol{v}) > G(\rho^\star; S_{\mathrm{KL}}^\star, \boldsymbol{v}) \quad \Leftrightarrow \quad \frac{A - v_j \exp(-r_j/\rho^\star)}{B - v_j} < \frac{A}{B} \quad \Leftrightarrow \quad \exp(-r_j/\rho^\star) > \frac{A}{B}.$$

The condition in (31) implies that $\exp(-r_j/\rho^\star) > A/B$, which means that $G(\rho^\star; S_{\mathrm{KL}}^\star \setminus \{j\}, \boldsymbol{v})$. Therefore, $S_{\mathrm{KL}}^\star \setminus \{j\}$ yields a higher objective value than $S_{\mathrm{KL}}^\star$ for the same $\rho^\star$, which contradicts the optimality of $S_{\mathrm{KL}}^\star$.

By calculation, we can obtain that $G(\rho^\star; S_{\mathrm{KL}}^\star \setminus \{j\}, \boldsymbol{v}) > G(\rho^\star; S_{\mathrm{KL}}^\star, \boldsymbol{v})$, which contradicts (31). Thus we must have $r_j \ge R_{\mathrm{KL}}^\delta(S_{\mathrm{KL}}^\star | \boldsymbol{v}) + \delta \rho^\star.$ $\qquad \square$

## C.3. Sub-Optimality Gap under Chi-square ambiguity Set

***Proof of Theorem 4.7.*** For the Chi-square ambiguity set, given $\boldsymbol{\theta}^\star$, we adopt the following shorthand notation for convenience: $p(\cdot|S) := p(\cdot|S, \boldsymbol{\theta}^\star)$, $\boldsymbol{v} := \exp(\boldsymbol{X}^\top \boldsymbol{\theta}^\star)$, $S^\star := S^\star_{\chi^2}$ and $\hat{S} := \hat{S}_{\chi^2}$.

**Bounding Term (I).** Recall that the dual formulation under Chi-square ambiguity set as shown in Lemma 3.4 is

$$R^\delta_{\chi^2}(S|\boldsymbol{v}) = \max_{0 \leq \eta \leq \frac{\sqrt{1+\delta}}{\sqrt{1+\delta}-1}} \left\{ \eta - \sqrt{(1+\delta) \sum_{i \in S_+} p(i|S)(\eta - r_i)^2} \right\}.$$

Let $\eta_{\max} = \frac{\sqrt{1+\delta}}{\sqrt{1+\delta}-1}$. To bound term (I), by applying the property $\max f - \max g \leq \max(f - g)$ and the inequality $\sqrt{a} - \sqrt{b} \leq \sqrt{|a - b|}$ for $a, b \geq 0$, we have:

$$R^\delta_{\chi^2}(S^\star|\boldsymbol{v}) - R^\delta_{\chi^2}(S^\star|\boldsymbol{v}^{\mathrm{LCB}}) \leq \max_{\eta \in [0, \eta_{\max}]} \left\{ \sqrt{(1+\delta) \sum_{i \in S^\star_+} \hat{p}(i|S^\star)(\eta - r_i)^2} - \sqrt{(1+\delta) \sum_{i \in S^\star_+} p(i|S^\star)(\eta - r_i)^2} \right\}$$

$$\leq \sqrt{1+\delta} \cdot \max_{\eta \in [0, \eta_{\max}]} \left\{ \sqrt{\sum_{i \in S^\star_+} |p(i|S^\star) - \hat{p}(i|S^\star)|(\eta - r_i)^2} \right\}$$

$$\leq \sqrt{1+\delta} \cdot \eta_{\max} \sqrt{\sum_{i \in S^\star_+} |p(i|S^\star) - \hat{p}(i|S^\star)|}$$

where the last inequality follows from the fact that $(\eta - r_i)^2 \leq \eta^2_{\max}$ since $r_i \in [0, 1]$ and $\eta_{\max} > 1$.

We decompose the sum inside the square root into two sets: $J_1 := \{i \in S^\star_+ : p(i|S^\star) \geq \hat{p}(i|S^\star)\}$ and $J_2 := \{i \in S^\star_+ : p(i|S^\star) < \hat{p}(i|S^\star)\}$. By Lemma C.1, we bound the aggregate probability deviation for both $J_1$ and $J_2$. For $k \in \{1, 2\}$, we have

$$\sum_{i \in J_k} (p(i|S^\star) - \hat{p}(i|S^\star)) \leq \sum_{i \in S^\star} p(i|S^\star)|u_i - u^{\mathrm{LCB}}_i| + \frac{3}{2} \max_{i \in S^\star} (u_i - u^{\mathrm{LCB}}_i)^2.$$

Combining these, term (I) is bounded as

$$R^\delta_{\chi^2}(S^\star|\boldsymbol{v}) - R^\delta_{\chi^2}(S^\star|\boldsymbol{v}^{\mathrm{LCB}}) \leq \frac{1+\delta}{\sqrt{1+\delta}-1} \sqrt{2 \sum_{i \in S^\star} p(i|S^\star)|u_i - u^{\mathrm{LCB}}_i| + 3 \max_{i \in S^\star} (u_i - u^{\mathrm{LCB}}_i)^2} \tag{32}$$

**Bounding Term (II).** Now we prove that Term (II) is non-positive. By Lemma 3.4,

$$R^\delta_{\chi^2}(\hat{S}|\boldsymbol{v}^{\mathrm{LCB}}) - R^\delta_{\chi^2}(\hat{S}|\boldsymbol{v})$$

$$= \max_{0 \leq \eta \leq \frac{\sqrt{1+\delta}}{\sqrt{1+\delta}-1}} \left\{ \eta - \sqrt{(1+\delta)\mathbb{E}_{i \sim \hat{p}(\cdot|\hat{S})}[(\eta - r_i)^2]} \right\} - \max_{0 \leq \eta \leq \frac{\sqrt{1+\delta}}{\sqrt{1+\delta}-1}} \left\{ \eta - \sqrt{(1+\delta)\mathbb{E}_{i \sim p(\cdot|\hat{S})}[(\eta - r_i)^2]} \right\}$$

$$\leq \sqrt{(1+\delta)\mathbb{E}_{i \sim p(\cdot|\hat{S})}[(\hat{\eta}^\star - r_i)^2]} - \sqrt{(1+\delta)\mathbb{E}_{i \sim \hat{p}(\cdot|\hat{S})}[(\hat{\eta}^\star - r_i)^2]},$$

where $\hat{\eta}^\star = \mathrm{argmax}_{0 \leq \eta \leq \frac{\sqrt{1+\delta}}{\sqrt{1+\delta}-1}} \left\{ \eta - \sqrt{(1+\delta)\mathbb{E}_{i \sim \hat{p}(\cdot|\hat{S})}[(\eta - r_i)^2]} \right\}$ is the optimizer for the LCB-based robust reward.

To show that this difference is non-positive, it suffices to prove

$$\mathbb{E}_{i \sim p(\cdot|\hat{S})}[(\hat{\eta}^\star - r_i)^2] \leq \mathbb{E}_{i \sim \hat{p}(\cdot|\hat{S})}[(\hat{\eta}^\star - r_i)^2],$$

that is

$$\frac{\sum_{i \in \hat{S}_+} v_i(\eta^\star - r_i)^2}{\sum_{i \in \hat{S}_+} v_i} \leq \frac{\sum_{i \in \hat{S}_+} v^{\mathrm{LCB}}_i(\eta^\star - r_i)^2}{\sum_{i \in \hat{S}_+} v^{\mathrm{LCB}}_i}.$$

By Lemma C.5, any item $j$ belongs to assortment $\hat{S}$ must satisfy

$$\sqrt{1+\delta} \cdot r_j \geq R^\delta_{\chi^2}(\hat{S}|\boldsymbol{v}^{\mathrm{LCB}}) + (\sqrt{1+\delta}-1)\hat{\eta}^\star$$
$$= -\sqrt{(1+\delta)\mathbb{E}_{i\sim\hat{p}(\cdot|\hat{S})}[(\hat{\eta}^\star - r_i)^2]} + \sqrt{1+\delta}\cdot\hat{\eta}^\star,$$

which is equivalent to

$$(\hat{\eta}^\star - r_j)^2 \leq \frac{\sum_{i\in\hat{S}_+} v_i^{\mathrm{LCB}}(\hat{\eta}^\star - r_i)^2}{\sum_{i\in\hat{S}_+} v_i^{\mathrm{LCB}}}.$$

Using the inequality above, we have

$$\frac{\sum_{i\in\hat{S}_+} v_i(\hat{\eta}^\star - r_i)^2}{\sum_{i\in\hat{S}_+} v_i} = \frac{\sum_{i\in\hat{S}_+} v_i^{\mathrm{LCB}}(\hat{\eta}^\star - r_i)^2 + \sum_{i\in\hat{S}_+}(v_i - v_i^{\mathrm{LCB}})(\hat{\eta}^\star - r_i)^2}{\sum_{i\in\hat{S}_+} v_i^{\mathrm{LCB}} + \sum_{i\in\hat{S}_+} v_i - v_i^{\mathrm{LCB}}}$$
$$\leq \frac{\sum_{i\in\hat{S}_+} v_i^{\mathrm{LCB}}(\hat{\eta}^\star - r_i)^2 + \sum_{i\in\hat{S}_+}(v_i - v_i^{\mathrm{LCB}})\frac{\sum_{i\in\hat{S}} v_i^{\mathrm{LCB}}(\hat{\eta}^\star - r_i)^2}{\sum_{i\in\hat{S}} v_i^{\mathrm{LCB}}}}{\sum_{i\in\hat{S}_+} v_i^{\mathrm{LCB}} + \sum_{i\in\hat{S}_+} v_i - v_i^{\mathrm{LCB}}}$$
$$= \frac{\sum_{i\in\hat{S}_+} v_i^{\mathrm{LCB}}(\hat{\eta}^\star - r_i)^2}{\sum_{i\in\hat{S}_+} v_i^{\mathrm{LCB}}},$$

where the last equality follows from the equation that $(a + c\cdot(a/b))/(b+c) = a/b$.

Consequently, we establish that

$$R^\delta_{\chi^2}(\hat{S}|\boldsymbol{v}^{\mathrm{LCB}}) - R^\delta_{\chi^2}(\hat{S}|\boldsymbol{v}) \leq 0. \tag{33}$$

**Final Bound.** Hence, by combining the upper bound for term (I) from Equation (32) and the non-positivity of term (II) from Equation (33), and substituting the utility gap from Equation (16), we obtain that

$$\mathrm{SubOpt}^\delta_{\chi^2}(\hat{S}|\boldsymbol{\theta}^\star) \leq \frac{1+\delta}{\sqrt{1+\delta}-1}\cdot$$
$$\sqrt{64\sum_{i\in S^\star} p(i|S^\star)\|\boldsymbol{x}_i\|_{\boldsymbol{H}^\lambda_T(\boldsymbol{\theta}^\star)^{-1}}\left(\sqrt{\log(N/\zeta)} + \sqrt{\lambda}W\right) + 3072\max_{i\in S^\star}\|\boldsymbol{x}_i\|^2_{\boldsymbol{H}^\lambda_T(\boldsymbol{\theta}^\star)^{-1}}\left(\sqrt{\log(N/\zeta)} + \sqrt{\lambda}W\right)^2}.$$

$\square$

**Lemma C.5.** *Consider the robust revenue under Chi-square ambiguity set with a fixed attraction vector $\boldsymbol{v}$. If an item $j\in[N]$ belongs to the optimal robust assortment $S^\star_{\chi^2}$, then its reward $r_j$ must satisfy:*

$$\sqrt{1+\delta}\cdot r_j \geq R^\delta_{\chi^2}(S^\star_{\chi^2}|\boldsymbol{v}) + (\sqrt{1+\delta}-1)\eta^\star, \quad (S^\star_{\chi^2}, \eta^\star) \in \underset{S\in\mathcal{S}_K, \eta\in[0,1/\delta]}{\mathrm{argmax}} H(\eta; S, \boldsymbol{v}) \text{ with } j\in S^\star_{\chi^2}.$$

*where $H(\eta; S, \boldsymbol{v}) := \eta - \sqrt{(1+\delta)\mathbb{E}_{i\sim p(\cdot|S,\boldsymbol{v})}[(\eta - r_i)^2]}$.*

*Proof of Lemma C.5.* We prove it by contradiction. Suppose there exists an item $j$ such that $j\in S^\star_{\chi^2}$ for some $(S^\star_{\chi^2}, \eta^\star)$ satisfying $(S^\star_{\chi^2}, \eta^\star)\in\mathrm{argmax}_{S\in\mathcal{S}_K, \eta\in[0,1/\delta]} H(\eta; S, \boldsymbol{v})$, but

$$\sqrt{1+\delta}\cdot r_j < R^\delta_{\chi^2}(S^\star_{\chi^2}|\boldsymbol{v}) + (\sqrt{1+\delta}-1)\eta^\star \quad\Leftrightarrow\quad (\eta^\star - r_j)^2 > \frac{\sum_{i\in S^\star_{\chi^2}} v_i(r_i - \eta^\star)^2}{\sum_{i\in S^\star_{\chi^2}} v_i}. \tag{34}$$

Then considering the new assortment $S^\star_{\chi^2}\setminus\{j\}\in\mathcal{S}_K$. To compare $H(\eta^\star; S^\star_{\chi^2}\setminus\{j\})$ with $H(\eta^\star; S^\star_{\chi^2})$, let $A := \sum_{i\in S^\star_{\chi^2}} v_i(r_i - \eta^\star)^2$, $B := \sum_{i\in S^\star_{\chi^2}} v_i$. Note that

$$H(\eta^\star; S^\star_{\chi^2}\setminus\{j\}, \boldsymbol{v}) > H(\eta^\star; S^\star_{\chi^2}, \boldsymbol{v}) \quad\Leftrightarrow\quad \frac{A - v_j(r_j - \eta^\star)^2}{B - v_j} < \frac{A}{B} \quad\Leftrightarrow\quad (r_j - \eta^\star)^2 > \frac{A}{B}.$$

The condition in (34) implies that $(r_j - \eta^\star)^2 > A/B$, which means $H(\eta^\star; S_{\chi^2}^\star \setminus \{j\}, \boldsymbol{v}) > H(\eta^\star; S_{\chi^2}^\star, \boldsymbol{v})$. This contradicts the optimality of $S_{\chi^2}^\star$ for the given $\eta^\star$.

Thus, we must have $\sqrt{1+\delta} \cdot r_j \geq R_{\chi^2}^\delta(S_{\chi^2}^\star|\boldsymbol{v}) + (\sqrt{1+\delta} - 1)\eta^\star$. $\qquad\square$

# D. Numerical Studies

We complement our theoretical results with synthetic experiments. The studies validate (i) the computational scaling of our algorithms (Appendix D.1), (ii) the unimodality assumption underlying Algorithm 5 (Appendix D.2), (iii) the value of robustness under distribution shift (Appendix D.3), (iv) the complementary roles of the three divergences (Appendix D.4), and (v) resilience under non-MNL model misspecification (Appendix D.5).

## D.1. Computational Performance

**Setup.** We evaluate the wall-clock runtime of our three robust algorithms across a range of inventory sizes. For each $N \in \{50, 100, 200, 500, 1000\}$, we generate contextual MNL instances satisfying Assumption 2.1 with $\boldsymbol{x}_i \in \mathbb{R}^{10}$ and $\|\boldsymbol{x}_i\|_2 \leq 1$, $\boldsymbol{\theta}^\star \in \mathbb{R}^{10}$ with $\|\boldsymbol{\theta}^\star\|_2 = 1$, $r_i \sim \text{Uniform}[0,1]$, and $v_i = \exp(\boldsymbol{x}_i^\top \boldsymbol{\theta}^\star)$. We fix $K = 10$, $\delta = 0.1$, and numerical tolerance $\varepsilon = 10^{-6}$. Table 2 reports the mean runtime (in seconds) $\pm$ one standard deviation over 10 independent random instances.

| $N$ | TV | KL | $\chi^2$ |
|---|---|---|---|
| 50 | $0.0008 \pm 0.0001$ | $0.0147 \pm 0.0068$ | $0.0006 \pm 0.0000$ |
| 100 | $0.0021 \pm 0.0001$ | $0.0313 \pm 0.0038$ | $0.0009 \pm 0.0001$ |
| 200 | $0.0072 \pm 0.0003$ | $0.0856 \pm 0.0052$ | $0.0018 \pm 0.0001$ |
| 500 | $0.0551 \pm 0.0057$ | $0.4745 \pm 0.0558$ | $0.0057 \pm 0.0007$ |
| 1000 | $0.2230 \pm 0.0069$ | $1.6222 \pm 0.1048$ | $0.0137 \pm 0.0037$ |

*Table 2.* Wall-clock runtime (seconds, mean $\pm$ s.d. over 10 instances) of the three robust algorithms.

**Result.** The empirical log–log slopes match the predicted complexities. TV exhibits a slope of approximately $2.08$, matching the $O(N^2 \log N)$ implementation-level complexity. KL achieves a slope of $1.83$, aligning with $\widetilde{O}(N^2)$. The $\chi^2$ algorithm has a slope of $1.19$, consistent with $O(N \log N \cdot \log(1/\varepsilon))$.

## D.2. Empirical Validation of Unimodality

Algorithm 5 (the $\chi^2$-robust solver) employs Golden Section Search (GSS) over the dual variable $\eta \in [0, \frac{\sqrt{1+\delta}}{\sqrt{1+\delta}-1}]$. This procedure attains $\varepsilon$-optimality only when the objective $H^\star(\eta)$ is unimodal. We empirically validate this assumption.

**Setup.** We generate 500 random instances with $N = 10$, $K = 3$, $r_i \sim \text{Uniform}[0,1]$, $v_i \sim \text{Uniform}[0.5, 2]$, and check unimodality of $H^\star(\eta)$ across ambiguity radii $\delta \in [0, 0.05]$.

**Result.** Table 3 reports the fraction of instances on which $H^\star(\eta)$ is unimodal. Unimodality holds on over $90\%$ of instances throughout the range $\delta \in [0, 0.05]$, supporting the use of GSS as a fast practical heuristic in the small-$\delta$ regime.

| $\delta$ | 0.01 | 0.02 | 0.05 |
|---|---|---|---|
| % unimodal | 93.4 | 93.4 | 90.0 |

*Table 3.* Fraction of 500 random instances ($N = 10$, $K = 3$) on which $H^\star(\eta)$ is unimodal, across varying ambiguity radii $\delta$.

Even on instances where $H^\star(\eta)$ fails to be unimodal, an $\varepsilon$-optimal solution can always be obtained by a global one-dimensional search such as uniform grid search over the dual interval $[0, \eta_{\max}]$.

### D.3. Out-of-Sample Robustness Under Distribution Shift

**Setup.** We use $N = 10$, $K = 3$, $r_i \sim \mathrm{Uniform}[0, 1]$, $v_i \sim \mathrm{Uniform}[0.5, 2]$, over 100 trials. For each trial, we compute the non-robust optimum $S^\star_{\mathrm{nom}}$ and the three robust policies $S^\star_{\mathrm{TV}}, S^\star_{\mathrm{KL}}, S^\star_{\chi^2}$. To simulate distribution shift, we generate a perturbed attraction vector $v'$ by multiplying the top-$\lfloor N/3 \rfloor$ highest-revenue items' attractions by a drop factor $\gamma$, with small additional multiplicative Gaussian noise; smaller $\gamma$ corresponds to a more severe shift. We then evaluate the out-of-sample revenue $R(S \mid v')$ of each policy under $v'$, varying the ambiguity radius over $\delta \in \{0.05, 0.1, 0.2, 0.3, 0.5\}$ and the drop factor over $\gamma \in \{0.1, 0.3, 0.5\}$.

**Result.** Figure 2 shows two findings. (i) All three robust policies consistently outperform the non-robust baseline across all shift severities, demonstrating its fragility under shift. (ii) The advantage of robustness increases with the uncertainty radius $\delta$, since larger ambiguity sets yield more protective assortments.

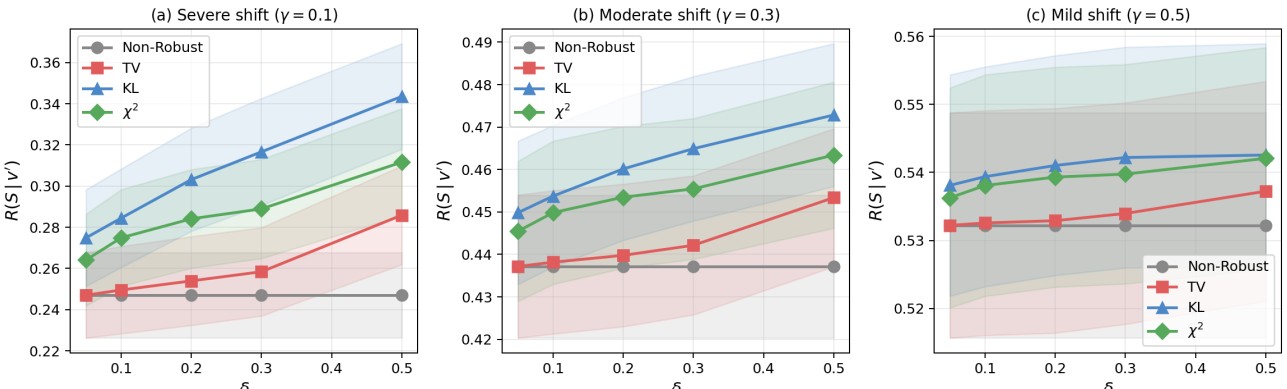

*Figure 2.* Out-of-sample revenue $R(S \mid v')$ vs. uncertainty radius $\delta$ under severe ($\gamma = 0.1$), moderate ($\gamma = 0.3$), and mild ($\gamma = 0.5$) distribution shift, comparing the non-robust baseline against the three robust policies. Shaded bands are $\pm 2$ standard errors over 100 trials ($N = 10$, $K = 3$).

### D.4. Comparison of Divergences Under Out-of-Ball Shifts

We compare the three robust policies when the realized shift lies *outside* all three ambiguity sets centered at the nominal model.

**Setup.** We use the same instance setup as in Appendix D.3 with $\delta = 0.2$ and 200 trials, and retain only those perturbed instances satisfying $D_{\mathrm{TV}}(q', p) > \delta$, $D_{\mathrm{KL}}(q', p) > \delta$, and $D_{\chi^2}(q', p) > \delta$ simultaneously, where $q' = p(\cdot \mid S^\star_{\mathrm{nom}}, v')$ and $p = p(\cdot \mid S^\star_{\mathrm{nom}}, v)$. This ensures the realized shift lies outside all three divergence balls. For each accepted pair $(v, v')$, we report both the raw out-of-sample revenue $R(S \mid v')$ and the revenue ratio

$$\mathrm{Ratio}(S) := \frac{R(S \mid v')}{R(S^\star_{\mathrm{nom}} \mid v')},$$

where $\mathrm{Ratio}(S) > 1$ means the robust policy outperforms the non-robust baseline in the perturbed environment. We sweep the drop factor $\gamma \in \{0.3, 0.2, 0.1\}$, with smaller $\gamma$ corresponding to a more severe shift.

**Result.** Figure 3 shows two findings. (i) All three robust policies achieve $\mathrm{Ratio} > 1$, indicating that robustness improves out-of-sample revenue even when the realized shift lies outside all three divergence balls. (ii) The relative advantage of robust policies grows as the shift becomes stronger (smaller $\gamma$); KL and $\chi^2$ in particular exhibit substantial gains over the non-robust baseline at $\gamma = 0.1$.

### D.5. Robustness Under Model Misspecification

We evaluate our robust MNL policies in a setting where the underlying choice model is non-MNL.

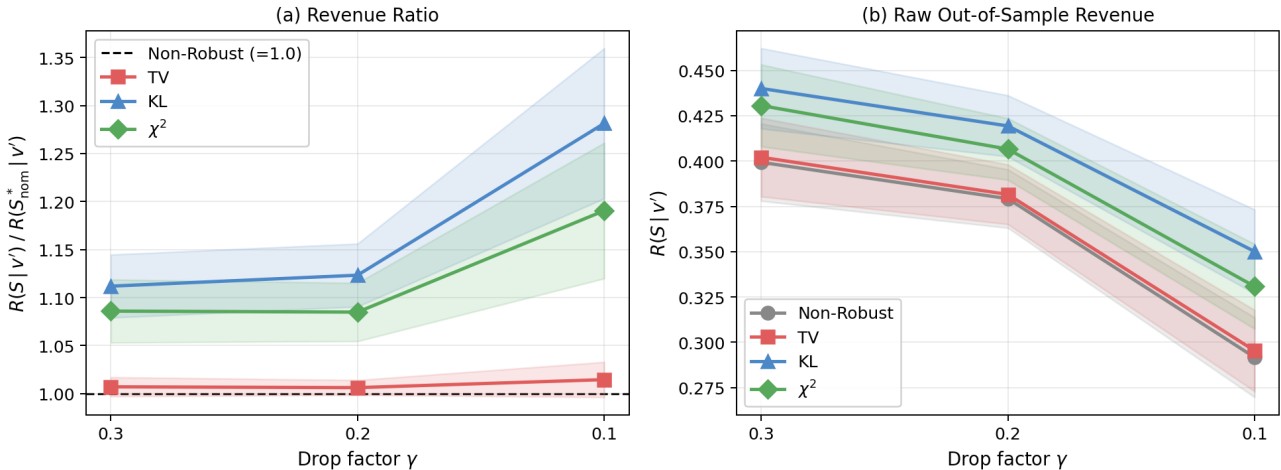

*Figure 3.* (a) Revenue ratio $R(S \mid \boldsymbol{v}')/R(S_{\mathrm{nom}}^{\star} \mid \boldsymbol{v}')$ and (b) raw out-of-sample revenue $R(S \mid \boldsymbol{v}')$ vs. drop factor $\gamma$, restricted to shifts that lie outside all three divergence balls. Smaller $\gamma$ corresponds to a more severe shift on the top-$\lfloor N/3 \rfloor$ revenue items. $N = 10$, $K = 3$, $\delta = 0.2$, 200 trials. Shaded bands are $\pm 2$ standard errors.

**Setup.** The true customer choice model is a 3-type Mixed MNL, so neither MNL nor Nested Logit (NL) is correctly specified. We compare our three robust MNL policies against a nominal two-nest NL baseline learned from the same data. For the three robust MNL methods, we estimate attraction values $\hat{\boldsymbol{v}}$ via MLE and apply Algorithms 1, 2, and 5 with $\delta = 0.15$. For the NL baseline, we fit by MLE with grid search over the dissimilarity parameter $\mu$ and optimize the fitted NL model. All learned policies are evaluated on the true Mixed MNL environment.

| $T$ | MNL-TV | MNL-KL | MNL-$\chi^2$ | NL nominal | Oracle |
|---|---|---|---|---|---|
| 50 | 0.519 | 0.554 | 0.553 | 0.449 | 0.564 |
| 200 | 0.518 | 0.557 | 0.553 | 0.449 | 0.564 |
| 800 | 0.519 | 0.557 | 0.554 | 0.449 | 0.564 |

*Table 4.* Out-of-sample expected revenue under a true 3-type Mixed MNL environment, comparing robust MNL policies against a Nested Logit baseline. "Oracle" is the optimal expected revenue under the true model.

**Result.** All three robust MNL methods outperform the NL baseline, indicating that the robust approach remains effective even under model misspecification. The robust policy guards against a range of nearby choice distributions, making it less sensitive to model errors. Hence the robust MNL methods remain competitive in a non-MNL environment.

