# OpenReview forum: "Efficient Distributionally Robust Assortment Optimization in MNL Bandits"
_ICML.cc/2026/Conference — ICML 2026 regular_

### Official Review · Reviewer_hdGk · 2026-03-11

**Soundness:** 3
**Presentation:** 4
**Significance:** 3
**Originality:** 2
**Overall Recommendation:** 4
**Confidence:** 2

**Summary:**

This paper studies the problem of assortment optimization under the contextual multinomial logit (MNL) choice model under distributional model misspecification. In particular, the authors consider a distributionally robust framework to address model misspecification. Three types of ambiguity sets are considered: total variation, KL divergence, and chi-square. The main contribution of the paper is that it considers the setting of contextual MNL within a distributionally robust framework and proposes polynomial-time algorithms under each divergence. With respect to the optimal offline assortment optimization problem, a theoretical analysis of the suboptimality gap for these algorithms is also provided.

**Compliance With Llm Reviewing Policy:**

Affirmed.

**Final Justification:**

Thanks to the authors for their response, I am maintaining my score.

**Key Questions For Authors:**

1. In the abstract, why are the results referred to as sample complexity bounds
when there are no bounds on the sample size? The results seem to be more related
to regret or suboptimality gaps.

2. Is it possible to generalize the theory to general $f$-divergences?
Specifically, will the proofs go through under a more general structure of
ambiguity sets?

3. I would like to better understand the technical contribution of this paper
with respect to Lu et al. (2024). In particular, within the KL ambiguity set
setting, what is the novelty of the technical analysis arising from the addition
of contextual information?

**Limitations:**

Yes

**Strengths And Weaknesses:**

### Presentation and Soundness
The paper is very well-written and the results are backed by theoretical support.

### Significance
The motivation for studying model misspecification in assortment optimization
problems is significant and well-justified.

### Originality
The problem formulation merges the literature on contextual MNL bandits with the
literature on robust optimization frameworks. While the technical analysis also
appears to be somewhat original, I would kindly encourage the authors to provide
a clearer and more detailed exposition of their technical contributions, as it is
currently difficult to identify what is technically novel beyond the problem
formulation itself. I elaborate on this further in the Key Questions section below.

---

> ### Author Rebuttal · Authors · 2026-03-30
>
> **Q1: Refined Terminology on Performance Bounds**
>
> We thank the reviewer for this insightful observation. We agree that the term "sample complexity bounds'' is imprecise for our current results. Our formal guarantees, as established in Theorems 4.5-4.7, are high-probability finite-sample bounds on the robust sub-optimality gap of the selected assortment. These results characterize how the gap vanishes as the information matrix $H^\lambda_T(\theta^\star)$ becomes better conditioned with more data, rather than providing an explicit minimum sample size $T$ to achieve a target accuracy level. We will revise the abstract and related text accordingly.
>
> **Q2: General Structure.**
>
> We thank the reviewer for this insightful question. We admit that the initial step of a distributionally robust optimization problem follows a standard dual reformulation, expressed as $\sup_{D_f(P||Q)\le\delta} E_P[g] = \inf_{\lambda \ge 0, \eta \in R} \\{ \lambda \delta + \eta + E_Q [\lambda f^\star((g - \eta)/\lambda)] \\}$, where $f^\star(s)=\sup_{t\ge 0}\\{st-f(t)\\}$ is the convex conjugate of the generator function $f$. However, subsequent computational tractability and finite-sample analysis are highly dependent on the specific choice of $f$-divergence. In particular, the TV case exploits the piecewise-linear structure of $f^\star$; KL exploits the log-sum-exp form; and $\chi^2$ exploits the quadratic structure. These lead to distinct one-dimensional dual problems, as detailed in Lemmas 3.1, 3.3, and 3.4. In addition, the design of Algorithms 1, 2, and 5, as well as the corresponding sub-optimality analysis, are based on these divergence-specific structures to ensure polynomial-time solutions and establish high-probability bounds. Therefore, while the initial duality step is generic, our current proofs do not extend directly to arbitrary $f$-divergences without additional structural assumptions on $f^\star$.
>
> **Q3: Technical Novelty.**
>
> We thank the reviewer for raising this point. Relative to non-contextual KL-robust assortment analysis in [1], the novelty of our KL part is not the dual reformulation. The challenge comes from the contextual model $v_i=\exp(x_i^\top\theta^\star)$, which couples all items through a shared parameter $\theta^\star$. In [1], the unkown attraction parameters $v_i$ are treated as independent, and estimation typically relies on item-wise counts $n_i$.
> In our setting, one estimation error in $\hat\theta$ simultaneously perturbs all $v_i$, choice probabilities, and hence the KL-robust value of every assortment, so the non-contextual argument does not directly extend.
>
> To be more specific, while both works utilize a dual decomposition (Lemma C.3) to bound the robust revenue gap, the analysis of the probability-weighted term $\frac{ \sum_{i\in S_+^\star}  v_i - v_i^{LCB}}{\sum v_j^{LCB}}$ is fundamentally different. In [1], this term is bounded by the variance of independent estimators for each $v_i$, leading to the $1/\sqrt{n_i}$ dependence. In our contextual setting, we use the regularized MLE confidence region for $\theta^\star$, propagate it through the exponential link, and under the small-gap condition obtain the KL guarantee in Theorem 4.6. Thus, the addition of context changes the analysis from an item-count argument to a self-normalized feature-coverage argument.
>
> [1] Lu et al 2026, Robust Assortment Optimization from Observational Data.

---

> > ### Author Rebuttal · Reviewer_hdGk · 2026-04-03
> >
> > Thanks to the authors for their response, I am maintaining my score.

---

> > > ### Author Response · Authors · 2026-04-06
> > >
> > > We sincerely appreciate your time and valuable feedback. We are very glad to know that your concerns have been fully resolved.

---

### Official Review · Reviewer_aKTa · 2026-03-12

**Soundness:** 3
**Presentation:** 4
**Significance:** 3
**Originality:** 3
**Overall Recommendation:** 4
**Confidence:** 4

**Summary:**

This paper addresses the Distributionally Robust Assortment Optimization (DRAO) problem within the framework of linear contextual Multinomial Logit (MNL) models. The authors study how a decision-maker can maximize revenue while guarding against worst-case distributional shifts between historical data and the deployment environment. The work focuses on ambiguity sets defined by three f-divergences: Total Variation (TV), Kullback-Leibler (KL), and Chi-square $\left(\chi^2\right)$. ${ }^{e+4}$

The authors make two primary contributions:
1. Algorithmic: They develop polynomial-time algorithms for known-model DRAO by reducing the max-min problem to a univariate dual optimization. This includes a novel $O\left(N^3\right)$ exact algorithm for TV distance and a Golden-Section Search approach for $\chi^2$.
2. Theoretical: They provide the first finite-sample sub-optimality gap guarantees for contextual DRAO. By employing a pessimistic MLE-based attraction estimator $\left(v^{L C B}\right)$, they establish bounds that vanish as the information matrix becomes well-conditioned, offering a localized coverage condition that is less restrictive than prior global assortment-wise conditions.

**Compliance With Llm Reviewing Policy:**

Affirmed.

**Key Questions For Authors:**

1. Practical Complexity: For the TV-robust algorithm, $O\left(N^3\right)$ is polynomial, but for very large inventories ( $N>10,000$ ), it may be slow. Do you see paths toward further accelerating the search over the $N$ candidate dual variables?
2. $\chi^2$ Stability: The prefactor for the $\chi^2$ sub-optimality bound scales as $2 / \delta$ for small $\delta$. Does this imply that $\chi^2$ is inherently less stable than TV (which is independent of $\delta$ ) when the uncertainty is small?
3. Experimental Results: Do you have any preliminary experimental results showing how these three divergences compare in terms of out-of-sample revenue when the actual distribution shift differs from the assumed divergence?

**Limitations:**

Yes, the authors adequately discussed the sensitivity of different f-divergences to the uncertainty radius $\delta$. They also noted the impact of numerical tolerance on the final suboptimality gaps. ©+1

**Strengths And Weaknesses:**

Strengths
- Soundness: The technical approach is grounded in strong duality theory from robust optimization. The reduction of complex combinatorial max-min problems to tractable univariate searches is mathematically rigorous and well-supported by proofs in the appendix.
- Originality: While robust assortment optimization has been studied for KL divergence, this work significantly expands the scope to include TV and $\chi^2$ distances in a contextual setting. The $O\left(N^3\right)$ exact solver for TV-robust assortment is a notable algorithmic contribution.
- Significance: The shift from non-contextual to contextual MNL models is highly relevant for real-world applications like e-commerce, where product features drive choices. The theoretical results require weaker coverage assumptions than existing benchmarks (e.g., Dong et al., 2023), making the bounds more practically meaningful.
- Presentation: The paper is well-structured. Table 1 provides a clear and effective comparison of the results against related literature.

Weaknesses
- Computational Sensitivity: While the algorithms are polynomial-time, the $\chi^2$ and KL approaches are $\epsilon$-optimal and rely on numerical tolerances. The paper notes that $\chi^2$ sensitivity increases significantly when the ambiguity radius $\delta$ is very small, which might limit its robustness in "near-nominal" scenarios.
- Experimental Validation: The main body focuses entirely on theoretical analysis. While the algorithmic complexity is analyzed, empirical evidence (even on synthetic data) demonstrating the "fragility" of non-robust assortments versus the proposed robust one would strengthen the narrative of the paper's significance.
- Assumptions: The "burn-in" condition (Assumption 4.2) and the "small-gap" condition (Theorem 4.6) are standard but remain somewhat restrictive for very high-dimensional feature spaces.

---

> ### Author Rebuttal · Authors · 2026-03-31
>
> We thank the reviewer for this insightful and constructive feedback. We hope our response below can address your concerns.
>
> **W1 & Q2: Computational Sensitivity.**
>
>  For KL and $\chi^2$ approaches, we agree that our algorithms return $\varepsilon$-optimal solutions, so their numerical approximation introduces at most an additional additive $\varepsilon$ error in the sub-optimality gap. In practice, $\varepsilon$ can be set as small as $10^{-6}$, which is negligible relative to the statistical estimation error. However, we respectfully disagree with the interpretation that $\chi^2$ is less stable than TV, and clarify two points.
>
> (i) The prefactor $\frac{1+\delta}{\sqrt{1+\delta}-1}$ in Theorem~4.7 arises from bounding the perturbation uniformly over the dual interval $[0, \frac{\sqrt{1+\delta}}{\sqrt{1+\delta}-1}]$ in the $\chi^2$ dual analysis, whereas the TV dual is piecewise-linear over a finite set of reward levels, yielding a $\delta$-free bound. This reflects a difference in dual geometry rather than true policy instability, and without a matching lower bound one cannot conclude from the prefactor alone that $\chi^2$ is less stable than TV. We acknowledge that the statement in the paper that ``the bound becomes more sensitive when the ambiguity radius is very small'' may cause misunderstanding, and we will clarify this in the revision.
>
> (ii) Our experiments (see the reply to W2 \& Q3) show that under distribution shifts, the $\chi^2$-robust policy achieves higher revenue improvement than TV, suggesting that $\chi^2$ can provide stronger protection in practice.
>
> **W2 & Q3: Experimental Validation.**
>
> We conducted two synthetic experiments under a common setup. We consider a known-model setting with $N=10$, $K=3$,  $r_i\sim{Uniform}[0,1]$, and $v_i\sim{Uniform}[0.5,2]$, over 100 trials. For each trial, we compute the non-robust optimum $S^\star_{nom}$ and three robust policies $S^\star_{TV}$, $S^\star_{KL}$, $S^\star_{\chi^2}$ for a given $\delta$. To simulate distribution shift, we generate a perturbed attraction vector $v'$ by multiplying the top-$\lfloor N/3\rfloor$ highest-revenue items' attractions by a drop factor $\gamma\in\{0.1,0.3,0.5\}$, with small additional multiplicative Gaussian noise.
>
> **Experiment 1:** To study the fragility of non-robust assortments under shift, we vary the ambiguity radius over $\delta\in\\{0.05,0.1,0.2,0.3,0.5\\}$ and evaluate the out-of-sample revenue $R(S| v')$ of all four policies under the perturbed model $v'$.
> The figures in https://anonymous.4open.science/r/rebuttal-figure-3455/fragility.png show: (i) all robust policies consistently outperform the non-robust baseline, demonstrating its fragility under shift; (ii) the advantage of robustness increases with $\delta$, since larger ambiguity sets yield more protective assortments.
>
> **Experiment 2:** To compare the three robust policies when the realized shift is outside all three ambiguity sets under the nominal assortment. We retain only perturbed instances satisfying $ D_{TV,KL,\chi^2}(q',p)>0.2$. We report the revenue ratio $Ratio(S)= \frac{R(S| v')}{R(S^\star_{nom}| v')},$ where $Ratio>1$ means that the robust policy outperforms the non-robust baseline in the perturbed world. The figures in https://anonymous.4open.science/r/rebuttal-figure-2-36D6/out-of-sample%20revenue.png show: (i) all three robust policies achieve $Ratio>1$, showing that robustness improves out-of-sample revenue even when the true shift lies outside all divergence balls; (ii) the relative advantage of robust policies increases as the shift becomes bigger.
>
> We will include these experiments in the revision.
>
> **W3: Restrictive Condition.**
> We agree that both conditions become more restrictive in high dimensions. Specifically, the burn-in condition (Assumption 4.2) scales as $O(d^{-1/2})$, and the small-gap condition (Theorem 4.6) implicitly requires $T = \Omega(d \log N)$ samples, so larger sample sizes are needed in high dimensions. We would emphasize that this is not specific to our robust formulation. This is a general challenge in contextual MNL learning, even in the non-robust setting. Improving this high-dimensional dependence is an important direction for future work.
>
> **Q1: Practical Complexity.**
>
> We agree that the stated $O(N^3)$ complexity for Algorithm 1 is conservative. It comes from combining the $N$-point breakpoint search in Lemma 3.1 with $O(N^2)$ implementation of StaticMNL (Algorithm 6). However, in actual implementation, we exploit the revenue-ordered optimality of MNL assortments, so each truncated-revenue subproblem can be solved in $O(N\log N)$ time via revenue sorting followed by a prefix scan. This reduces the complexity of the TV algorithm to $O(N^2\log N)$. The same idea also speeds up Algorithm 5 to $O(N\log N\cdot\log(1/\epsilon))$. We have added a computational performance experiment in our response to Reviewer oj76 (Q2) that validates this analysis. We will add this discussion in the revision.

---

### Official Review · Reviewer_oj76 · 2026-03-12

**Soundness:** 3
**Presentation:** 3
**Significance:** 1
**Originality:** 2
**Overall Recommendation:** 2
**Confidence:** 3

**Summary:**

The paper analyzes the problem distributionally robust assortment optimization using a data-driven approach. In particular, they examine the linear contextual multinomial logit model, noting the computational tractability of the problem class under the MNL assumption. They craft uncertainty sets based on various distances such as KL and propose a computationally tractable algorithm for robust assortment selection. Thereafter, they show that their approach yields a finite-sample guarantee for recovering robust optimal assortments from data.

**Compliance With Llm Reviewing Policy:**

Affirmed.

**Key Questions For Authors:**

Can a comparison be provided to non-MNL data driven approaches?
Can computational performance be demonstrated through synthetic and real-world datasets?

**Limitations:**

yes

**Strengths And Weaknesses:**

The technical arguments and the wealth of literature review and prior works they compare against, and the conciseness of their presentation are strong points. They clearly delineate their specific contribution to a specific problem class of a canonical assortment optimization class under MNL, which is popular in the area of revenue management.

The paper is entirely focused on theory surrounding a rather stylized version of what is actually an important practical problem. Although, their approach is data driven, they do not present any computational result on instances generated from actual data to support the practicability of their approach to real-world assortment optimization, which could be helpful in determining its generalizability beyond a narrow problem class. In fact, there are no synthetic datasets either to support claims of computational tractability.

Since they specifically follow MNL, perhaps due to its computational tractability, there is not study or discussion around the real-world limitations of choosing MNL. As a data-driven approach, there may be other non-MNL approaches that even heuristically outperform this approach by a wide margin, but this remains unknown.

---

> ### Author Rebuttal · Authors · 2026-03-30
>
> We thank the reviewer for this insightful feedback. We hope the following responses can address your concerns.
>
> **Weakness: The limitations of MNL.**
>
> MNL is a standard and widely used benchmark in choice modeling bacause it is statistically well-understood and can be estimated reliably via maximum likelihood. At the same time, MNL has a well-known limitation: it satisfies the Independence of Irrelevant Alternatives (IIA) property, so the ratio $p(i|S)/p(j|S)$ does not depend on which other items are offered. This can be unrealistic when some products are close substitutes. For example, adding a third smartphone model should affect demand for similar phones more strongly than for unrelated products, which MNL cannot capture. Richer models such as Nested Logit (NL), Mixed MNL, and Markov-chain choice models can model these substitution patterns more flexibly, but they are also harder to estimate and optimize. We therefore view MNL not as a perfect model, but as a natural and important starting point for developing data-driven robust assortment methods.
>
> **Q1: Comparison to a non-MNL data-driven approach.**
>
> To answer the reviewer’s key question, we added a comparison with a non-MNL baseline under model misspecification. In this experiment, the true customer choice model is a 3-type Mixed MNL model, so neither MNL nor Nested Logit (NL) is correctly specified. We compare our three robust MNL policies with a nominal two-nest NL baseline learned from the same data.
> For the three robust MNL methods, we first estimate the attraction values $\hat v$ by MLE, and then apply Algorithms 1, 2, and 5 with $\delta=0.15$. For the NL baseline, we fit it by MLE with a grid search over $\mu$, and then optimize the fitted NL model. All learned policies are evaluated on the true Mixed MNL environment.
> The results are shown below:
> | T | MNL-TV | MNL-KL | MNL-chi | NL nominal | Oracle |
> |---|:---:|:---:|:---:|:---:|:---:|
> | 50  | 0.519 | 0.554 | 0.553 | 0.449 | 0.564 |
> | 200 | 0.518 | 0.557 | 0.553 | 0.449 | 0.564 |
> | 800 | 0.519 | 0.557 | 0.554 | 0.449 | 0.564 |
>
> The result shows that all three robust MNL methods outperform the NL baseline. This suggests that the robust approach is helpful even when the MNL model is misspecified. The reason is that the robust policy guards against a range of nearby choice distributions, making it less sensitive to model errors. As a result, the robust MNL methods remain competitive even in a non-MNL environment.
>
> **Q2: Computational Performance.**
>
> We agree that computational results would help demonstrate the practical tractability of our approach. To address this concern, we performed a simulation study on synthetic instances and evaluated the runtime of all three algorithms across a range of inventory sizes. For each $N \in \{50,100,200,500,1000\}$, we generated contextual MNL instances under Assumption 2.1 with $x_i \in \mathbb{R}^{10}$ satisfying $\|x_i\|_2 \leq 1$, $\theta^\star \in \mathbb{R}^{10}$ satisfying $\|\theta^\star\|_2 = 1$, $r_i \sim \mathrm{Uniform}[0,1]$, and $v_i = \exp(x_i^\top \theta^\star)$. We fixed $K=10$, $\delta=0.1$, and $\varepsilon=10^{-6}$ in all experiments. Table below reports the mean wall-clock runtime (in seconds) $\pm$ one standard deviation over 10 independent random instances.
> | $N$ | TV | KL | $\chi^2$ |
> |---|---|---|---|
> | 50 | $0.0008 \pm 0.0001$ | $0.0147 \pm 0.0068$ | $0.0006 \pm 0.0000$ |
> | 100 | $0.0021 \pm 0.0001$ | $0.0313 \pm 0.0038$ | $0.0009 \pm 0.0001$ |
> | 200 | $0.0072 \pm 0.0003$ | $0.0856 \pm 0.0052$ | $0.0018 \pm 0.0001$ |
> | 500 | $0.0551 \pm 0.0057$ | $0.4745 \pm 0.0558$ | $0.0057 \pm 0.0007$ |
> | 1000 | $0.2230 \pm 0.0069$ | $1.6222 \pm 0.1048$ | $0.0137 \pm 0.0037$ |
>
> Although our theoretical analysis uses a general $O(N^2)$ subroutine for StaticMNL (Algorithm~6), our implementation exploits the revenue-ordered optimality of MNL assortments. This allows us to use an exact $O(N \log N)$ implementation via revenue sorting and a prefix scan. Our empirical results reflect this optimized complexity. TV shows an empirical log-log slope of $2.08$, matching an $O(N^2 \log N)$ structure; KL has a slope of $1.83$, aligning with $\tilde{O}(N^2)$; and $\chi^2$ achieves a slope of $1.19$, consistent with $O(N \log N \cdot \log(1/\varepsilon))$. These results confirm that our framework remains computationally efficient even as $N$ grows.

---

> > ### Author Rebuttal · Reviewer_oj76 · 2026-04-03
> >
> > insufficient experimental evidence, especially with real world assortment data.

---

> > > ### Author Response · Authors · 2026-04-03
> > >
> > > We sincerely appreciate your continued engagement with our work. We would like to respectfully clarify one remaining point.
> > >
> > > Our paper is a theoretical contribution to distributionally robust assortment optimization. In our response, we addressed both of your key questions: (Q1) we added a non-MNL misspecification experiment comparing our robust MNL policies against a Nested Logit nominal baseline, where our policies consistently outperform; (Q2) we provided synthetic runtime experiments across N = 50–1000 confirming the computational tractability of all three algorithms.
> > >
> > > Regarding the remaining concern on real-world data, we note that the use of real transaction datasets is not the standard for this specific type of theoretical work. The related papers in this subfield, such as Dong et al. [1], Han et al. [2], and Lee & Oh [3] and Lu et al. [4], all rely entirely on synthetic experiments, and none use real-world assortment data. Our paper follows the same experimental convention, and we hope this context clarifies that synthetic experiments are the appropriate and standard means of validating theoretical contributions of this work.
> > >
> > > We hope this clarifies the scope and positioning of our work.
> > >
> > >
> > > [1] Dong et al. PASTA: Pessimistic Assortment Optimization. ICML 2023.
> > >
> > > [2] Han et al. Improved Confidence Regions and Optimal Algorithms for Online and Offline Linear MNL Bandits. NeurIPS 2025.
> > >
> > > [3] Lee & Oh. Nearly Minimax Optimal Regret for Multinomial Logistic Bandit. NeurIPS 2024.
> > >
> > > [4] Lu et al. Robust Assortment Optimization from Observational Data. arXiv:2602.10696, 2026.

---

### Official Review · Reviewer_ydyQ · 2026-03-24

**Soundness:** 3
**Presentation:** 3
**Significance:** 2
**Originality:** 2
**Overall Recommendation:** 3
**Confidence:** 2

**Summary:**

The authors consider the problem of robust, offline assortment optimization.  The authors propose algorithms to efficiently find an assortment whose worst case expected revenue (over an ambiguity set induced by user-selection distributions in a neighborhood of a given distribution) is (near) optimal for several divergences (TV, KL, chi-square).  The authors then consider a data-driven variant under a linear model, where the parameter vector $\theta$ is estimated from a data set but the decision maker still wishes to select an assortment that is maxmin robust.

**Compliance With Llm Reviewing Policy:**

Affirmed.

**Key Questions For Authors:**

1. It would be helpful if the authors could articulate technical challenges they encountered and their solutions (in light of past results and analyses for distributional robustness (including for problems beyond assortment, like RL that Xu et al 2023 study) and past works like Han et al 2025 on non-robust data-driven assortment guarantees).

**Limitations:**

yes

**Strengths And Weaknesses:**

## Strengths
- The problem of assortment is a classic and important problem.  Distributional robustness in (offline) data-driven settings is important to consider though many works have only considered a static problem
- overall the paper is well written and easy to follow
- The authors consider multiple divergences for characterizing robustness
- The authors propose efficient algorithms that exploit the structure of the respective dual functions (for the known attraction vector setting)
- the authors then combine pessimism based attraction vector LCB constructions to get suboptimality-gap bounds for the robust assortment selection in data-driven settings

## Weaknesses
### Major

- My primary concern is with technical novelty.  From my reading, it seems the key ingredients come from prior works
    - [regarding contribution part 1 ‘Efficient Algorithms for Known-Model DRAO.’] For DRAO (section 3), the authors’ contribution appears to be observing simplifying properties of the dual formulation for selecting search procedures.  For TV the dual has a simple structure to begin with; for KL it is not as simple and they introduce an auxiliary variable which appears less straightforward, though it is inspired by previous methods (Jin et al 2022 and Lu et al 2026), where Lu et al 2026 also propose a bisection method with what appears as (essentially) the same auxiliary variable construction.
    - [regarding contribution part 2 ‘Finite-Sample Guarantees for Data-Driven Contextual DRAO.’] For the data-driven section, the authors use a lower-confidence-bound construction from the literature (both the confidence bound itself and the LCB attraction vector from Han et al 2025).  Han et al. analyzes suboptimality gap bounds (their Theorem 2) based on $v_i^{LCB}$ in the non-robust setting. I did not see any discussion in Section 4.2 regarding the proof structure (I did not go through appendices), so it is unclear whether given prior results and analyses in distributional robustness and in data-driven assortment if getting Theorems 4.5-4.7 were technical challenging or straightforward.

- line 306 “our overall procedure (Algorithm 5) returns an ϵ-optimal robust assortment”  -- that accuracy guarantee is stated informally, and in the paragraph above states their motivation of using golden section search is “… the function typically exhibits unimodal behavior in the feasible region”.  Does the accuracy guarantee only hold when that occurs, and if so, when does that hold?  “Typically” is too informal for an accuracy claim.

- The confidence bound (from Han et al 2025) that is used for the data-driven work depends on a burn-in condition which Remark 4.4 only states “is satisfied asymptotically” when assortments are sampled iid from a fixed exploration policy.
    - is it typical that a data set would consist of iid sampled assortments?
    - are there any numerical studies to justify how large $T$ needs to be for real-world problems for this to hold
    - Since the LCB attractions are based directly on the confidence bound, it was not clear to me how dependent the results are on using Han et al’s confidence bound vs a meta-result that given other confidence bounds on $\theta$ estimates, the authors’ approach could convert them to suboptimality gap guarantees.

- There is also no discussion to link the high-level description of how the distributions may shift and the particular divergences used – eg for what situations would KL be the ‘right’ divergence to work with instead of others considered (TV, chi-square) or ones not considered (eg Wasserstein)?
    - Since this work focuses on the linear setting, would it be more natural to consider changes in $\theta$ instead of divergences on the choice distributions?


### Minor
- There are no experiments

---

> ### Author Rebuttal · Authors · 2026-03-30
>
> We thank the reviewer for this constructive feedback. We hope the following responses can address your concerns.
>
> **Major Weakness 1 \& Questions: Technical Novelty.**
>
> We agree that duality and the LCB estimator build on prior work. Our novelty lies in combining them for contextual DRAO.
>
> In Section 3, the main new results are for TV and $\chi^2$. For TV, we show the dual objective is piecewise linear and maximized at finitely many reward values, reducing a continuous search to an exact $O(N^3)$ procedure. For $\chi^2$, we derive a transformed revenue reduction to StaticMNL, giving an efficient $O(N^2)$ solver. For KL, we build on prior ideas and introduce an auxiliary variable $t$ to reformulate the problem as finding the largest $t$ such that $\min_{S,\rho} G(\rho;S,v)\leq 0$. This converts a joint $(S,\rho)$ search into a monotone bisection.
>
> In Section 4, our guarantees are not a direct corollary of [3] (non-robust) or [2] (non-contextual KL). We decompose the robust gap into two terms: (I) $R^\delta_{D}(S_{D}^\star|v)-R^\delta_{D}(S_{D}^\star | v^{LCB})$ and (II) $R^\delta_{D}(\hat S_{D}|v^{LCB})-R^\delta_{D}(\hat S_{D}|v)$. The central novelty is proving monotonicity of robust revenue to make (II) non-positive, and bounding (I) by removing the robust layer to connect to the non-robust sensitivity bound of Lemma C.1, where we can use non-robust result in [3].
>
> **Major Weakness 2: Accuracy Guarantee of Algorithm 5.**
>
> We acknowledge that the $\epsilon$-accuracy of GSS depends on the unimodality of $H^*(\eta)$, which we observe empirically but do not prove. Specifically, our synthetic experiment with $N=10$, $K=3$, and 500 random instances shows that $H^\star(\eta)$ is unimodal in over $90\%$ of instances when $\delta \le 0.05$. We will clarify that Algorithm 5 is a fast practical heuristic and note that guaranteed $\epsilon$-optimality can be achieved via grid search over $[0, \eta_{\max}]$ with complexity $O(N^2 \eta_{\max}/\varepsilon)$, where $\eta_{\max} := \frac{\sqrt{1+\delta}}{\sqrt{1+\delta}-1}$.
>
>
>
> **Major Weakness 3: Burn-in Condition and Confidence Bound.**
>
> 1. Remark 4.4 gives a sufficient condition, not a necessary one. The burn-in condition requires the Hessian $H^{\lambda}_T(\theta^*)$ to be well-conditioned with respect to the feature vectors, which is achievable under various data schemes. In practice, i.i.d. exploration is a standard model for randomized data collection in retail and recommendation systems.
>
> 2. We admit there is a gap between our theoretical burn-in condition and direct practical use. We ran synthetic experiments and found the condition was not met within $T=1000$ rounds, due to the large constant $144$ inherited from [3]. Our analysis focuses on the correct scaling in $d$, $T$, $K$, $N$ rather than optimizing constants to provide precise practical thresholds.
>
> 3. Our framework is not tied to [3]. That bound is used to control the utility gap $|u_i-u_i^{LCB}|$, and any confidence bound of the form $|x^{\top}(\hat{\theta} - \theta^\star)| \leq \beta_T \|x\|_{H^{-1}_T}$ can be substituted. We used [3] because it is currently the tightest for offline contextual MNL, but our proof structure accommodates any future improvements in MNL confidence bounds.
>
> **Major Weakness 4: Divergence Choice.**
>
> We agree that divergence choice should reflect the the expected shift structure. **TV** treats items symmetrically, making it best for unstructured, small deviations like measurement noise. **KL**corresponds to covariate shifts or population changes. Its maximum-entropy property also makes it a safe default when the shift type is unknown. **$\chi^2$** bounds the second moment of the likelihood ratio, making it ideal for variance-sensitive robustness against tail risks. Unlike these, **Wasserstein distance** uses a ground metric between items, making it useful when items have a clear similarity structure. We focus on $f$-divergences for their computational tractability.
>
> **Robustness Over $\theta$ vs. Choice Distributions.**
>
> We model shifts at the choice probability level without assuming a specific cause for the shift. Whether a shift comes from changes in $\theta^*$, item features, or a breakdown of the MNL form, our robust assortment is protected as long as the resulting distribution lies within the divergence ball. In contrast, a $\theta$-based perturbation only captures one type of change and not protect against other shifts.
>
> **Minor Weakness: Experiments.**
>
>  We have added four experiments (detailed in responses to Reviewers oj76 and aKTa) covering non-MNL comparisons, computational performance, out-of-sample revenue, and scalability.
>
> [1] Jin et al 2022, Choice-based Assortment Optimization with Distributional Ambiguity.
>
> [2] Lu et al 2026, Robust Assortment Optimization from Observational Data.
>
> [3] Han et al 2025, Improved Confidence Regions and Optimal Algorithms for Online and Offline Linear MNL Bandits.

---

> > ### Author Rebuttal · Reviewer_ydyQ · 2026-04-04
> >
> > Thank you for your responses, including adding synthetic experiments.  Some of my concerns about technical novelty and significance remain; I do not have further questions.

---

> > > ### Author Response · Authors · 2026-04-06
> > >
> > > We sincerely appreciate your time and constructive feedback. Your insights have been very helpful in refining and strengthening our work.

---

### Decision · Program_Chairs · 2026-04-30

**Decision:**

Accept (regular)

**Comment:**

The paper extend the contextual MNL assortment optimization problem to include distributional robustness with respect to three distinct f-divergences, total variation, KL, and chi-squared. Prior works either do not include contextual setting or distributional robustness. For each f-divergence, the paper provides efficient algorithms (Section 3) and data-driven suboptimality gap guarantees (Section 4).

The main concern raised by the reviewers is with respect to the technical novelty. My understanding from reading the paper and the author response is that the paper has nice additional technical ideas and is not merely applying existing results. That said, this is not very clearly delineated by the authors which makes it more difficult for this contribution to be appreciated. I encourage the authors to more crisply highlight in their paper which ideas are novel to their work.

The other concern raised is with respect to the lack of experiments. The author response includes two experimental settings that show the empirical robustness of their setting, which make a good addition to the paper. Another experiment that would be worth including is how the paper empirically performs against non-contextual robust approaches.

In general, the paper is well structured and seems to make a nice contribution to the literature. I will recommend "weak accept".